# Revisiting Differentiable Structure Learning: Inconsistency of $\ell_1$ Penalty and Beyond

## Abstract

Recent advances in differentiable structure learning have framed the combinatorial problem of learning directed acyclic graphs as a continuous optimization problem. Various aspects, including data standardization, have been studied to identify factors that influence the empirical performance of these methods. In this work, we investigate critical limitations in differentiable structure learning methods, focusing on settings where the true structure can be identified up to Markov equivalence classes, particularly in the linear Gaussian case. While Ng et al. (2024) highlighted potential non-convexity issues in this setting, we demonstrate and explain why the use of $\ell_1$-penalized likelihood in such cases is fundamentally inconsistent, even if the global optimum of the optimization problem can be found. To resolve this limitation, we develop a hybrid differentiable structure learning method based on $\ell_0$-penalized likelihood with hard acyclicity constraint, where the $\ell_0$ penalty can be approximated by different techniques including Gumbel-Softmax. Specifically, we first estimate the underlying moral graph, and use it to restrict the search space of the optimization problem, which helps alleviate the non-convexity issue. Experimental results show that the proposed method enhances empirical performance both before and after data standardization, providing a more reliable path for future advancements in differentiable structure learning, especially for learning Markov equivalence classes.

## 1 Introduction

Probabilistic graphical models, such as Bayesian networks, are powerful tools for capturing complex probabilistic relationships in a concise way (PEARL, 1988; Koller & Friedman, 2009). Their graph structures, usually encoded as Directed Acyclic Graphs (DAGs), allow efficient representation of data dependencies and have become essential in fields like health (Tennant et al., 2021) and economy (Awokuse & Bessler, 2003). Traditionally, learning these structures involves discrete methodologies. Constraint-based methods, which leverage conditional independence tests, are one common approach (Spirtes & Glymour, 1991; Spirtes et al., 2001). Another popular technique involves score-based methods, where the search space of potential graphs is explored based on scoring functions (Koivisto & Sood, 2004; Singh & Moore, 2005; Cussens, 2011; Yuan & Malone, 2013; Chickering, 2002; Peters & Bühlmann, 2014). Given the combinatorial nature of the task, greedy search strategies have been commonly used (Chickering, 1996; Chickering et al., 2004).

In recent years, Zheng et al. (2018) introduced a continuous formulation for characterizing the acyclicity constraint, effectively converting the discrete nature of the structure learning problem into one that can be approached using gradient-based optimization techniques. Although this formulation still involves nonconvex optimization, it opened the door to applying efficient gradient-based methods. This formulation has since inspired a wide range of extensions, being adapted to deal with nonlinearity (Yu et al., 2019; Lachapelle et al., 2019; Zheng et al., 2020; Ng et al., 2022b; Kalainathan et al., 2022), latent confounding (Bhattacharya et al., 2021; Bellot & van der Schaar, 2021), interventional data (Brouillard et al., 2020; Faria et al., 2022), time series data (Pamfil et al., 2020; Sun et al., 2021), and missing data (Wang et al., 2020; Gao et al., 2022). Other applications include multi-task learning (Chen et al., 2021), and working with federated learning systems (Ng & Zhang, 2022; Gao et al., 2021), domain adaptation (Yang et al., 2021), and recommendation system (Wang et al., 2022).

This move toward continuous structure learning has prompted growing attention to both its theoretical underpinnings and practical performance. Researchers like Wei et al. (2020) and Ng et al. (2022a) have investigated the optimality and convergence properties of continuous, constrained optimization techniques (Zheng et al., 2018). Meanwhile, Deng et al. (2023) provided insight into how an appropriately designed optimization scheme can reach the global minimum for least squares objectives in simple cases. Further refinements have also been proposed, with Zhang et al. (2022) and Bello et al. (2022) highlighting challenges such as gradient vanishing in existing DAG constraints (Zheng et al., 2018; Yu et al., 2019) and suggesting potential improvements.

Recently, Ng et al. (2024) highlighted the non-convexity issues in differentiable structure learning methods, particularly in the linear Gaussian setting where the true structure can be identified up to Markov equivalence classes. While non-convexity poses major challenges in this context, we further identify another critical issue: $\ell_1$-penalized likelihood is inconsistent, even if the global optimum of the optimization problem can be found. To address these limitations, we propose a method that resolves the $\ell_1$ inconsistency and enhances empirical performance, both before and after data standardization, even under non-convex conditions.

**Contributions** In this work, we tackle fundamental challenges in differentiable structure learning, particularly in the linear Gaussian case, by focusing on the limitations of penalized likelihood approaches. Our contributions include:

- We identify and demonstrate the inconsistency of using $\ell_1$-penalized likelihood in differentiable structure learning methods, even if the global optimum of the optimization problem can be found, particularly when learning Markov equivalence classes in the linear Gaussian case.

- We develop a differentiable structure learning method that optimizes an $\ell_0$-penalized likelihood with hard acyclicity constraints and incorporates a moral graph estimation procedure, where the $\ell_0$ penalty is approximated by differentiable techniques, such as Gumbel-Softmax. We call our method CALM (**C**ontinuous and **A**cyclicity-constrained **L**0-penalized likelihood with estimated **M**oral graph). CALM not only addresses $\ell_1$ inconsistency, but also results in a solution much closer to the true structure or its Markov equivalent graphs. Our method provides a more reliable path for future advancements in differentiable structure learning, especially for learning Markov equivalence classes.

- Our method performs consistently well both before and after data standardization, demonstrating its robustness.

## 2 BACKGROUND

In this section, we introduce our problem setting and, while reviewing the hard and soft DAG constraints, we also revisit NOTEARS (Zheng et al., 2018) and GOLEM (Ng et al., 2020).

### 2.1 PROBLEM SETTING

**Setup** In this work, we focus on linear Gaussian Structural Equation Models (SEMs), where the variables $X = (X_1, \ldots, X_d)$ follow linear relationships represented by a DAG. The model is expressed as $X = B^\top X + N$. Here, $B \in \mathbb{R}^{d \times d}$ is the weighted adjacency matrix encoding the relationships between variables. Specifically, an entry $B_{ij} \neq 0$ indicates a directed edge from $X_j$ to $X_i$. The noise vector $N = (N_1, \ldots, N_d)$ consists of independent noise terms, each corresponding to a variable $X_i$. The noise terms are assumed to follow a normal distribution with diagonal covariance matrix $\Omega = \text{diag}(\sigma_1^2, \ldots, \sigma_d^2)$, where $\sigma_i^2$ represents the variance of $N_i$. Given a DAG, a moral graph is an undirected graph obtained by removing the directions of the edges in the DAG and connecting all pairs of parents of common children.

Unlike many existing methods that assume equal noise variances (Zheng et al., 2018; Yu et al., 2019; Zhang et al., 2022; Bello et al., 2022), in this study, we focus on the general non-equal noise variance (NV) case, where the variances $\sigma_1^2, \ldots, \sigma_d^2$ are not assumed to be equal. Our goal is to estimate the DAG $G$ or its Markov equivalence class (MEC) from the data matrix $\mathbf{X} \in \mathbb{R}^{n \times d}$, consisting of $n$ i.i.d. samples drawn from the distribution $P(X)$.

## 2.2 Hard and Soft DAG Constraint

In the context of DAG learning, the DAG constraint, denoted as $h(B)$, ensures that the learned structure is a DAG when $h(B) = 0$. NOTEARS (Zheng et al., 2018) employs a hard DAG constraint, whereas GOLEM (Ng et al., 2020) introduces a soft DAG constraint.

**NOTEARS and hard DAG constraint**    NOTEARS solves the following constrained optimization problem

$$\min_{B \in \mathbb{R}^{d \times d}} \ell(B; \mathbf{X}) := \frac{1}{2n} \|\mathbf{X} - \mathbf{X}B\|_F^2 + \lambda \|B\|_1 \quad \text{subject to} \quad h(B) = 0.$$

Here, $\ell(B; X)$ is the least squares loss with $\ell_1$ penalty, and $h(B) = 0$ enforces the hard DAG constraint. The constrained optimization problem can be solved using standard algorithms such as augmented Lagrangian method (Wright, 2006), quadratic penalty method (Ng et al., 2022a), and barrier method (Bello et al., 2022).

**GOLEM and soft DAG constraint**    The GOLEM framework aims to maximize the likelihood of the observed data under the assumption of a linear Gaussian model. There are two formulations in GOLEM, one assuming equal noise variance across variables (GOLEM-EV), and the other allowing for non-equal noise variance (GOLEM-NV).

Unlike NOTEARS, GOLEM adopts the soft DAG constraint, making the problem unconstrained. In other words, GOLEM incorporates $h(B)$ as an additional penalty term in the score function, controlled by the hyperparameter $\lambda_2$. Here, we only review the the non-equal noise variance formulation, GOLEM-NV, which is the focus of this paper. Assuming that $\mathbf{X}$ is centered and that the sample covariance matrix $\Sigma = \frac{1}{n}\mathbf{X}^\top \mathbf{X}$, GOLEM-NV's unconstrained optimization problem is

$$\min_{B \in \mathbb{R}^{d \times d}} \mathcal{L}(B; \Sigma) + \lambda_1 \|B\|_1 + \lambda_2 h(B),$$

$$\text{where} \quad \mathcal{L}(B; \Sigma) = \frac{1}{2} \sum_{i=1}^{d} \log \left( \left( (I - B)^\top \Sigma (I - B) \right)_{i,i} \right) - \log |\det(I - B)|.$$

Here, $\mathcal{L}(B; \Sigma)$ is the likelihood function of linear Gaussian directed graphical models. This allows us to use $B$ and $\Sigma$ to express GOLEM-NV's formulation.

## 3 Inconsistency of $\ell_1$ penalty in Structure Learning

In this section, we explore the inconsistency of the $\ell_1$ penalty in structure learning by comparing its behavior with $\ell_0$ in linear Gaussian cases. We demonstrate this inconsistency through extensive experiments and conclude with a counterexample that highlights the issue.

### 3.1 $\ell_0$ vs $\ell_1$ penalty in Linear Gaussian Cases

In structure learning for linear Gaussian cases, score-based methods, specifically with the BIC score (Schwarz, 1978; Chickering, 2002), often aim to recover the sparsest underlying DAG that best explains the observed data (Singh & Moore, 2005; Cussens, 2011; Yuan & Malone, 2013; Chickering, 2002). In the asymptotic case where the population covariance matrix, denoted by $\Sigma^*$, is available, this can, loosely speaking, be formulated as

$$\min_{B, \Omega} \|B\|_0 \quad \text{subject to} \quad (I - B)^{-\top} \Omega (I - B)^{-1} = \Sigma^* \quad \text{and} \quad B \text{ is a DAG}.$$

Recall that $B$ is the weighted adjacency matrix representing the structure of the DAG and $\Omega$ is the diagonal matrix of noise variances. That is, the goal is to minimize the $\ell_0$ norm of $B$, i.e., the number of edges, while maximizing the likelihood by satisfying the covariance constraint and ensuring that $B$ is a valid DAG. In other words, the objective is to recover the sparsest DAG, $\hat{B}$, along with its corresponding $\hat{\Omega}$, that can generate the observed covariance matrix $\Sigma^*$ (i.e., $(I - \hat{B})^{-\top} \hat{\Omega} (I - \hat{B})^{-1} = \Sigma^*$). Under the sparsest Markov faithfulness assumption (Raskutti & Uhler, 2018), the estimated $\hat{B}$ will be Markov equivalent to the true graph $B^*$.

Many previous work, including GOLEM, replace the $\ell_0$ penalty with the more tractable $\ell_1$ penalty. The corresponding optimization problem becomes

$$\min_{B,\Omega} \|B\|_1 \quad \text{subject to} \quad (I-B)^{-\top}\Omega(I-B)^{-1} = \Sigma^* \quad \text{and} \quad B \text{ is a DAG.} \tag{1}$$

The $\ell_1$ penalty encourages smaller edge weights but introduces a key inconsistency: it doesn't guarantee true sparsity in the resulting structure. Minimizing the $\ell_1$ norm may lead to a denser structure with more edges than the solution from minimizing the $\ell_0$ norm. This occurs because $\ell_1$ favors edges with small absolute values, even if they represent spurious edges. In some cases, the sum of the absolute values of more edges can be smaller than that of fewer, larger-magnitude edges, which leads $\ell_1$-based methods to include unnecessary edges. As a result, the learned structure may deviate from the true DAG or its Markov equivalence class. Taking GOLEM as an example, even if the covariance constraint $(I-B)^{-\top}\Omega(I-B)^{-1} = \Sigma^*$ is satisfied in both $\ell_0$- and $\ell_1$-based formulations, the structural properties of the solution can differ significantly.

In Section 3.2, we demonstrate this with a large number of experiments, showing that a large proportion of DAGs $\tilde{B}$ satisfying the covariance constraint $(I-\tilde{B})^{-\top}\Omega(I-\tilde{B})^{-1} = \Sigma^*$ have smaller $\ell_1$ norms than the true graph $B^*$. Moreover, in these DAGs satisfying the covariance constraint and having smaller $\ell_1$ norms than the true graph $B^*$, we can always find ones with much larger $\ell_0$ values than $B^*$, meaning they do not correspond to the true graph $B^*$ or its Markov equivalence class, with the structural Hamming distance (SHD) computed over the true and estimated CPDAG (Completed Partially Directed Acyclic Graph), which we refer to as SHD of CPDAG in this paper, far from zero.

### 3.2 Experiment: Assessing the Inconsistency of $\ell_1$ penalty

In this section, we demonstrate and evaluate the inconsistency of the $\ell_1$ penalty in likelihood-based GOLEM through experiments. We generated 1,000 true DAGs $B^*$, compute their corresponding covariance matrices $\Sigma^*$ under infinite sample conditions. For each $B^*$, we use Cholesky decomposition to generate large number of DAGs $\tilde{B}$ that can generate the same $\Sigma^*$. Our goal is to identify the $\tilde{B}$ with the minimum $\ell_1$ norm among these $\tilde{B}$s, denoted as $B_{\ell_1}$, and compare it with $B^*$ in terms of $\ell_1$ norm, $\ell_0$ norm (i.e., edge count), and record its SHD of CPDAG. Additionally, we also record the proportion of $\tilde{B}$s that satisfy the covariance constraint (i.e., generate the same $\Sigma^*$) but have a smaller $\ell_1$ norm than $B^*$, to give an intuitive sense of the extent of $\ell_1$ inconsistency.

**True DAG generation and covariance matrix computation**  We generate 1000 8-node ($d=8$) ER1 graphs $B^*$s . The data is generated with a fixed noise ratio of 16, where the variances of two randomly selected noise variables are set to 1 and 16, respectively. The variances of the remaining noise variables are sampled uniformly from the range $[1, 16]$. The edge weights are uniformly sampled from $[-2, -0.5] \cup [0.5, 2]$. For each $B^*$, we compute the corresponding population covariance matrix $\Sigma^*$ under infinite samples using the equation $\Sigma^* = (I-B^*)^{-\top}\Omega^*(I-B^*)^{-1}$.

**Generating DAGs which meet covariance constraint**  Following the sparsest permutation approach developed by Raskutti & Uhler (2018), for each true covariance matrix $\Sigma^*$, we generate all possible $d!$ permutations of its rows and columns. For each permuted covariance matrix, we apply Cholesky decomposition to find a DAG $\tilde{B}$ that generates the permuted covariance matrix. After that, we restore $\tilde{B}$ to the original variable order. This ensures that all $\tilde{B}$ satisfies the covariance constraint $(I-\tilde{B})^{-\top}\tilde{\Omega}(I-\tilde{B})^{-1} = \Sigma^*$, while remaining a valid DAG. After the above steps, for each of the 1,000 true $B^*$, we identified $d!$ different DAGs $\tilde{B}$ that all generate $\Sigma^*$.

**Metrics and analysis**  For each true DAG $B^*$, we analyze the following metrics across all $d!$ DAGs $\tilde{B}$ that satisfy the covariance constraint: (1) $\ell_1$ norm comparison: we calculate the $\ell_1$ norm of each $\tilde{B}$ and record the proportion of $\tilde{B}$s whose $\ell_1$ norm is smaller than that of $B^*$; (2) selecting $\tilde{B}$ with the minimum $\ell_1$ norm: among the $d!$ $\tilde{B}$s, we select the one with the smallest $\ell_1$ norm, denoted as $B_{\ell_1}$. We then compare $B_{\ell_1}$ with $B^*$ in terms of $\ell_1$, and record its edge count and SHD of CPDAG (to test its distance to $B^*$ or its Markov equivalence class).

**Experimental results**  After running experiments for 1000 $B^*$s, we summarized the result in Table 1. Table 1 shows a comparison between the true DAG $B^*$ and the $B_{\ell_1}$ that generate the same

covariance matrix $\Sigma^*$. On average, for each true DAG $B^*$, 77.86% of the $d!$ DAGs $\tilde{B}$ satisfying the covariance constraint have smaller $\ell_1$ norms than $B^*$. In the 1,000 runnings, the average $\ell_1$ norm of $B_{\ell_1}$ is 4.22, which is smaller than the average $\ell_1$ norm of $B^*$, which is 10.04. The average $\ell_0$ norm (number of edges) of $B_{\ell_1}$ is 22.74, which is larger than the $\ell_0$ norm (number of edges) of $B^*$, which is 8. The average SHD of CPDAG between $B_{\ell_1}$ and $B^*$ is 19.97. In addition, in each running, $B_{\ell_1}$ consistently has a smaller $\ell_1$ norm than $B^*$, a larger $\ell_0$ norm (number of edges), and a SHD of CPDAG greater than zero, indicating that $B_{\ell_1}$ is structurally different from $B^*$ and its Markov equivalence class. These results demonstrate two key points: (1) a significant proportion of DAGs $\tilde{B}$ that satisfy the covariance constraint have smaller $\ell_1$ norms than $B^*$, and (2) in each running, we can find a counterexample (i.e., the $B_{\ell_1}$) where the $\ell_1$ norm is smaller, but the $\ell_0$ norm is larger, and the resulting DAG is not equivalent to $B^*$ or its Markov equivalence class. This supports the conclusion that $\ell_1$-based solutions are inconsistent in recovering the true structure.

Table 1: Comparison of $\tilde{B}$s which generate $\Sigma^*$ with True DAG $B^*$. The results are averaged over 1,000 simulated $B^*$s. The "Proportion" column reflects the average percentage of DAGs $\tilde{B}$ with $\ell_1$ norms smaller than that of $B^*$ among $d!$ $\tilde{B}$s per $B^*$.

| Metric | $B^*$ | $B_{\ell_1}$ | Proportion of $\tilde{B}$ with $\|\tilde{B}\|_1 < \|B^*\|_1$ |
|---|---|---|---|
| Average $\ell_1$ norm | $10.04 \pm 0.04$ | $4.22 \pm 0.03$ | $77.86\% \pm 0.46\%$ |
| Average $\ell_0$ norm (Number of Edges) | $8.0 \pm 0.0$ | $22.74 \pm 0.15$ | |
| Average SHD of CPDAG | $0 \pm 0.0$ | $19.97 \pm 0.17$ | |

### 3.3 CASE STUDY: A SPECIFIC COUNTEREXAMPLE

In this section, we present a 3-node counterexample to demonstrate the inconsistency of the $\ell_1$ penalty in differentiable structural learning. Specifically, we compare a true weighted adjacency matrix $B^*$ with an estimated adjacency matrix $\tilde{B}$, and show that although both matrices can generate the same covariance matrix, their $\ell_0$ norm (edge count), $\ell_1$ norm, and structural differences, measured by SHD of CPDAG, reveal the inconsistency of the $\ell_1$ penalty.

The true adjacency matrix $B^*$ and its corresponding noise covariance matrices $\Omega^*$ are given as:

$$B^* = \begin{bmatrix} 0 & \frac{1}{2} & 0 \\ 0 & 0 & 0 \\ 0 & -1 & 0 \end{bmatrix}, \quad \Omega^* = \begin{bmatrix} 16 & 0 & 0 \\ 0 & 4 & 0 \\ 0 & 0 & 1 \end{bmatrix}.$$

The estimated adjacency matrix $\tilde{B}$ and its corresponding noise covariance matrices $\tilde{\Omega}$ are:

$$\tilde{B} = \begin{bmatrix} 0 & \frac{1}{2} & \frac{1}{10} \\ 0 & 0 & -\frac{1}{5} \\ 0 & 0 & 0 \end{bmatrix}, \quad \tilde{\Omega} = \begin{bmatrix} 16 & 0 & 0 \\ 0 & 5 & 0 \\ 0 & 0 & \frac{4}{5} \end{bmatrix}.$$

Both matrices $B^*$ and $\tilde{B}$, along with their respective noise covariance matrices, generate the same covariance matrix:

$$\Sigma^* = \begin{bmatrix} 16 & 8 & 0 \\ 8 & 9 & -1 \\ 0 & -1 & 1 \end{bmatrix}.$$

We have $\|B^*\|_0 = 2$ and and $\|\tilde{B}\|_0 = 3$, indicating that $B^*$ is sparser than $\tilde{B}$. However, when considering the $\ell_1$ norm, we observe that: $\|B^*\|_1 = \frac{3}{2}$ and $\|\tilde{B}\|_1 = \frac{4}{5}$. That is, although $\tilde{B}$ has a higher $\ell_0$ norm, it achieves a lower $\ell_1$ norm, highlighting the inconsistency between the two norms. Therefore, the optimization problem in Eq. equation 1 may return $\tilde{B}$, which is clearly not Markov equivalent to $B^*$.

This counterexample demonstrates the inconsistency of the $\ell_1$ penalty: it may lead to solutions with smaller total edge weights (resulting in a lower $\ell_1$ norm), but these solutions may still have more edges (a higher $\ell_0$ norm) and deviate from the true DAG structure and its Markov equivalence class, even if these solutions can generate the same covariance matrix as the ground truth DAG.

# 4 CONTINUOUS AND ACYCLICITY-CONSTRAINED $\ell_0$-PENALIZED LIKELIHOOD WITH ESTIMATED MORAL GRAPH

In Section 2.2, we reviewed the GOLEM-NV formulation proposed by Ng et al. (2020), which aims to maximize the data likelihood utilizes an $\ell_1$ penalty and soft DAG constraint. We refer to this original model as GOLEM-NV-$\ell_1$ throughout this paper. The problem formulation can be expressed as

$$\min_{B \in \mathbb{R}^{d \times d}} \mathcal{L}(B; \Sigma) + \lambda_1 \|B\|_1 + \lambda_2 h(B).$$

However, as pointed out by Ng et al. (2024), GOLEM-NV-$\ell_1$ suffers from significant non-convexity, often leading to suboptimal local minima with poor performance, both before and after data standardization. Moreover, as we demonstrated in section 3, the $\ell_1$ penalty leads to inconsistent solutions. To address these limitations, we propose CALM (**C**ontinuous and **A**cyclicity-constrained **L**0-penalized likelihood with estimated **M**oral graph), a differentiable structure learning method that optimizes an $\ell_0$-penalized likelihood with hard DAG constraints and incorporates moral graphs. Our experiments demonstrate that CALM significantly improves performance compared to the original GOLEM-NV-$\ell_1$, yielding results much closer to the true DAG or its Markov equivalence class.

In the following subsections, we will first introduce the implementation of CALM, followed by the experimental setup and an evaluation of CALM's performance under various configurations. Finally, we will highlight the revisions and improvements in CALM over Existing Methods, emphasizing its practical design and robust performance compared to GOLEM-NV-$\ell_1$ and NOTEARS.

## 4.1 INTRODUCTION TO CALM

$\ell_0$ **penalty and its approximation with Gumbel Softmax**   CALM begins with applying an $\ell_0$ penalty to regularize the likelihood, enforcing sparsity in the learned adjacency matrix. Inspired by Ng et al. (2022b); Kalainathan et al. (2022), we use Gumbel Softmax as an example to show how we achieve the approximation of $\ell_0$ penalty in our approach, as it proved to be the most effective and robust $\ell_0$ approximation among those we experimented with in section 4.5. When using Gumbel Softmax (Jang et al., 2017) to approximate $\ell_0$ penalty, CALM starts by representing the learned adjacency matrix $B$ as an element-wise product of a learned mask, $g_\tau(U) \in \mathbb{R}^{d \times d}$, which determines the presence of edges, and a learned parameter matrix, $P \in \mathbb{R}^{d \times d}$, which learns the weights of the edges. The mask $g_\tau(U)$ is generated using the Gumbel-Softmax approach. Here, $U_{i,j}$ represents the logits, and a logistic noise $G_{i,j} \sim \text{Logistic}(0, 1)$ is added to $U_{i,j}$, producing $g_\tau(U_{i,j}) = \sigma((U_{i,j} + G_{i,j})/\tau)$, where $\tau$ is the temperature that controls the smoothness of the Softmax, and $\sigma(\cdot)$ is the logistic sigmoid function. As the optimization process proceeds, the values of $g_\tau(U_{i,j})$ approach either 0 or 1, approximating an $\ell_0$ penalty.

**Incorporating the moral graph and hard DAG constraints**   Furthermore, CALM incorporates a learned moral graph $M \in \{0, 1\}^{d \times d}$ to restrict the optimization to edges within the moral graph, thus reducing the search space. Note that similar idea has been used in various existing works such as Loh & Bühlmann (2014); Nazaret et al. (2024). This moral graph acts as a filter over the Gumbel-Softmax mask, allowing only edges present in the moral graph. The final learned $B$ can be represented as $B = M \circ g_\tau(U) \circ P$, incorporating both the sparsity from the Gumbel-Softmax mask and the structural constraints from the moral graph. Here, $B$ contains the edge weights for the DAG, and its structure is determined by the mask $M \circ g_\tau(U)$.

CALM's objective function, incorporating the Gumbel-Softmax mask, moral graph, and hard DAG constraints into the GOLEM-NV-$\ell_1$ formulation, is given by

$$\min_{U \in \mathbb{R}^{d \times d}, P \in \mathbb{R}^{d \times d}} \mathcal{L}(M \circ g_\tau(U) \circ P; \Sigma) + \lambda_1 \|M \circ g_\tau(U)\|_1 \quad \text{subject to} \quad h(M \circ g_\tau(U)) = 0.$$

where $\mathcal{L}(M \circ g_\tau(U) \circ P; \Sigma)$ is the likelihood term, and the $\lambda_1 \|M \circ g_\tau(U)\|_1$ term approximates the $\ell_0$ penalty for sparsity. Here, both the $\ell_0$ penalty for sparsity and the DAG constraints are applied to the final learned mask $M \circ g_\tau(U)$, which determines the presence of edges.

## 4.2 EXPERIMENTAL SETUP

Across all experiments in section 4, we simulate Erdös–Rényi graphs (ERDdS & R&wi, 1959) with $kd$ edges, denoted as ERk graphs, with edge weights uniformly sampled from $[-2, -0.5] \cup [0.5, 2]$. For all experiments, the data is generated with a fixed noise ratio of 16. Specifically, the variances of two randomly selected noise variables are set to 1 and 16, respectively, while the variances of the remaining noise variables are sampled uniformly from the range $[1, 16]$. This setting ensures a realistic variation in noise across the variables, aligning with the assumptions of non-equal noise variances (NV). Regarding CALM's optimization problem, we solve it using a quadratic penalty method inspired by Ng et al. (2022a), where each subproblem is tackled using gradient-based optimization with the Adam optimizer. The computational complexity per iteration is $O(d^3)$, which is comparable to most other differentiable structure learning methods in the linear case, such as NOTEARS and GOLEM. Regarding parameter tuning, we determined the hyperparameter $\lambda_1$, which controls the $\ell_0$-penalty, through extensive experiments. Various values such as 0.0005, 0.05, and 0.5 were tested, with 0.005 consistently yielding the best results across different settings. Other parameters were also carefully tuned to select the optimal ones. Further implementation details of our experiments in section 4 are in Appendix A. From this point forward, unless otherwise specified, we use "CALM" to refers to the specific version of the method where the $\ell_0$-penalty is approximated using Gumbel Softmax.

The following four sub-sections evaluate CALM's performance: first, we compare soft vs. hard DAG constraints and moral graph vs. no moral graph; second, we assess the effect of data standardization; third, we test different $\ell_0$ approximation methods. Finally, we compare CALM with baseline approaches. For each scenario, we conducted 10 experiments and calculated the mean SHD of CPDAG, precision of skeleton, recall of skeleton, and their standard errors.

## 4.3 IMPACT OF MORAL GRAPH AND SOFT/HARD DAG CONSTRAINTS

Recall that NOTEARS (Zheng et al., 2018) adopts a hard DAG constraint while GOLEM (Ng et al., 2020) uses a soft DAG constraint. Here, we evaluate the impact of incorporating the moral graph and using either soft or hard DAG constraints on the results of the $\ell_0$-penalized likelihood optimization. We consider linear Gaussian models with 50 variables and ER1 graphs. Here, we focus on the nonconvexity aspect of the optimization problem, and thus set the sample size to infinity to eliminate finite sample errors (the way we achieve infinite samples is in Appendix A.4). The experiment results for finite samples are included in Section 4.6. Furthermore, following Reisach et al. (2021); Kaiser & Sipos (2022), we standardize the data. All experiments were conducted with the Gumbel-Softmax approach to approximate $\ell_0$ penalty. The implementation details of Gumbel-Softmax-based $\ell_0$ penalty and the hard DAG constraints is in Appendix A.2. The implementation details of soft DAG constraints is in Appendix A.3.

Table 2: Impact of moral graph and soft/hard DAG constraints for 50-node ER1 graphs under data standardization

|  | SHD of CPDAG | Precision of Skeleton | Recall of Skeleton |
| --- | --- | --- | --- |
| Soft Constraints Without Moral | 33.8 ± 2.7 | 0.98 ± 0.01 | 0.43 ± 0.05 |
| Soft Constraints With Moral | 7.6 ± 2.5 | 0.98 ± 0.01 | 0.95 ± 0.03 |
| Hard Constraints Without Moral | 16.7 ± 3.4 | 0.88 ± 0.03 | 0.97 ± 0.01 |
| Hard Constraints With Moral (CALM) | 5.5 ± 1.9 | 0.98 ± 0.01 | 0.99 ± 0.00 |

**Comparison of soft and hard DAG constraints** From the results in Table 2, we observe that using hard DAG constraints leads to a lower SHD of CPDAG compared to soft DAG constraints, regardless of whether the moral graph is incorporated. This suggests that, even when both formulation with soft and hard DAG constraints converge to local optima, the hard DAG constraint results are closer to the true adjacency matrix $B$ or its Markov equivalence class.

One explanation for the improved performance of hard DAG constraints is the use of a quadratic penalty method (QPM) (Ng et al., 2022a). In this framework, the hyperparameter $\rho$, which controls the weight of the DAG constraint, is progressively increased during optimization, with each $\rho$ value triggering a full subproblem optimization. This leads to a more refined optimization process. In

contrast, the soft DAG constraint uses a fixed $\rho$, resulting in only one subiteration and possibly worse convergence. Additionally, hard constraints ensure that the final graph is always a DAG, eliminating the need for post-processing, whereas soft constraints often require post-processing to enforce acyclicity (Ng et al., 2020), which may introduce errors and increase SHD of CPDAG.

**Impact of including the moral graph.** Table 2 also shows that incorporating the moral graph improves performance in both soft and hard DAG constraint settings, with a notably lower SHD of CPDAG. The moral graph reduces the search space by focusing on edges within it, which is especially beneficial in higher-dimensional settings like our 50-node experiments, where the reduction in search space is more substantial. This significantly simplifies the optimization process and leads to better convergence towards the true adjacency matrix or its Markov equivalence class.

In summary, CALM, combining hard DAG constraints and the moral graph, delivers the best results.

### 4.4 IMPACT OF DATA STANDARDIZATION

Ng et al. (2024) previously pointed out that the original GOLEM-NV-$\ell_1$ formulation performed poorly both before and after data standardization. To further evaluate the robustness of CALM, we conduct experiments to compare its performance with (CALM-Standardized) and without data standardization (CALM-Non-Standardized). Here, we use infinite samples to eliminate finite sample error and consider a 50-node linear Gaussian model with ER1 graphs. The implementation details of Gumbel-Softmax-based $\ell_0$ penalty and the hard DAG constraints for CALM is in Appendix A.2. In Table 3, CALM shows consistently low SHD of CPDAG before and after data standardization, demonstrating its stability and robustness across both standardized and non-standardized data. Interestingly, this is in constrast with the observation by Reisach et al. (2021); Kaiser & Sipos (2022) that differentiable structure learning methods do not perform well after data standardization, which further validate the robustness of our method.

Table 3: Impact of Data Standardization on CALM for 50-node ER1 graphs

|  | SHD of CPDAG | Precision of Skeleton | Recall of Skeleton |
| --- | --- | --- | --- |
| CALM-Non-Standardized | 9.9 ± 3.4 | 0.95 ± 0.02 | 0.99 ± 0.01 |
| CALM-Standardized | 5.5 ± 1.9 | 0.98 ± 0.01 | 0.99 ± 0.00 |

### 4.5 COMPARISON OF $\ell_0$ APPROXIMATION METHODS

We compare our method with three $\ell_0$ approximations (CALM, CALM-STG, CALM-Tanh) and the original GOLEM-NV-$\ell_1$. For all methods here, We used infinite samples to eliminate finite sample error and considered 50- node linear Gaussian model with ER1 graphs. Additional results for ER4 with 50 nodes and ER1 with 100 nodes are presented in Appendix B. Here, CALM maintains our definition, specifically referring to our method that employs the Gumbel-Softmax approximation for the $\ell_0$ penalty. CALM-STG (Stochastic Gates) refers to our method that uses stochastic gates to approximate $\ell_0$ penalty (Yamada et al., 2020), while CALM-Tanh (Hyperbolic Tangent) refers to our method that employs the smooth hyperbolic tangent function to approximate $\ell_0$ penalty (Bhattacharya et al., 2021). The key parameter selection and implementation details for STG and Tanh in approximating the $\ell_0$ penalty can be found in Appendix A.5.

Table 4: Comparison of our method using different $\ell_0$ approximations and original GOLEM-NV-$\ell_1$ for 50-node ER1 graphs under both data standardization and no data standardization.

|  | Standardized | | | Non-Standardized | | |
| --- | --- | --- | --- | --- | --- | --- |
|  | SHD of CPDAG | Precision of Skeleton | Recall of Skeleton | SHD of CPDAG | Precision of Skeleton | Recall of Skeleton |
| CALM | 5.5 ± 1.9 | 0.98 ± 0.01 | 0.99 ± 0.00 | 9.9 ± 3.4 | 0.95 ± 0.02 | 0.99 ± 0.01 |
| CALM-STG | 5.9 ± 1.5 | 0.97 ± 0.01 | 0.99 ± 0.00 | 34.1 ± 3.6 | 0.79 ± 0.02 | 0.95 ± 0.01 |
| CALM-Tanh | 8.6 ± 1.6 | 0.95 ± 0.01 | 0.99 ± 0.01 | 51.7 ± 2.5 | 0.69 ± 0.02 | 0.91 ± 0.01 |
| GOLEM-NV-$\ell_1$ | 56.2 ± 3.5 | 0.60 ± 0.03 | 0.55 ± 0.05 | 121.1 ± 6.1 | 0.35 ± 0.02 | 0.76 ± 0.03 |

Table 4 shows that CALM-STG achieves competitive results after data standardization but underperforms without standardization. CALM-Tanh shows the weakest performance among the three $\ell_0$ approximations methods. Only CALM performs well both with and without data standardization,

highlighting its robustness, which is why we ultimately selected Gumbel-Softmax as the $\ell_0$ approximation method as our representative implementation. Additionally, all three methods outperform GOLEM-NV-$\ell_1$, underscoring the importance of $\ell_0$ penalty approximation, as $\ell_1$ penalty suffers from inconsistency (as discussed in Section 3). Moreover, this also shows that the effectiveness of combining $\ell_0$ approximation with moral graphs and hard constraints.

## 4.6 COMPARISON WITH BASELINE METHODS

We finally compare the performance of CALM against several baseline methods, including the original GOLEM-NV-$\ell_1$, NOTEARS, PC (Spirtes & Glymour, 1991), FGES (Ramsey et al., 2017) and DAGMA (Bello et al., 2022) (see Appendix A.6 for baseline methods' implemention details). We evaluated the methods at two different sample sizes: $n = 1000$ and $n = 10^6$. We considered a 50-node linear Gaussian model with ER1 and ER4 graphs, as well as a 100-node linear Gaussian model with ER1 graphs. Here, the moral graph in CALM is estimated from finite samples (see Appendix A.1 for how we estimate the moral graph). Specifically, for the 1000-sample experiments, the moral graph is estimated from 1000 samples, and for the $10^6$-sample experiments, the moral graph is estimated from $10^6$ samples. All results presented here are from experiments with standardized data. The additional experimental results for data without standardization are presented in Appendix C.

Table 5: Comparison with baseline methods for 50-node ER1, 50-node ER4, and 100-node ER1 graphs under data standardization, using 1000 and $10^6$ samples.

| | 50-node ER1 graphs | | | | | |
| --- | --- | --- | --- | --- | --- | --- |
| | 1000 Samples | | | $10^6$ Samples | | |
| | SHD of CPDAG | Precision of Skeleton | Recall of Skeleton | SHD of CPDAG | Precision of Skeleton | Recall of Skeleton |
| CALM | 12.1 ± 2.7 | 0.93 ± 0.01 | 0.98 ± 0.00 | 7.0 ± 2.5 | 0.97 ± 0.01 | 0.98 ± 0.01 |
| GOLEM-NV-$\ell_1$ | 60.0 ± 3.9 | 0.58 ± 0.03 | 0.65 ± 0.07 | 55.6 ± 2.6 | 0.60 ± 0.02 | 0.63 ± 0.07 |
| NOTEARS | 46.3 ± 1.9 | 0.76 ± 0.01 | 0.81 ± 0.02 | 46.6 ± 1.9 | 0.75 ± 0.02 | 0.79 ± 0.02 |
| PC | 11.0 ± 1.4 | 0.98 ± 0.01 | 0.92 ± 0.01 | 2.8 ± 0.8 | 0.99 ± 0.01 | 0.99 ± 0.00 |
| FGES | 8.4 ± 2.4 | 0.94 ± 0.02 | 0.98 ± 0.00 | 1.3 ± 0.9 | 1.00 ± 0.00 | 1.00 ± 0.00 |
| DAGMA | 73.3 ± 4.0 | 0.57 ± 0.02 | 0.95 ± 0.01 | 70.6 ± 3.4 | 0.59 ± 0.02 | 0.95 ± 0.01 |
| | 50-node ER4 graphs | | | | | |
| | 1000 Samples | | | $10^6$ Samples | | |
| | SHD of CPDAG | Precision of Skeleton | Recall of Skeleton | SHD of CPDAG | Precision of Skeleton | Recall of Skeleton |
| CALM | 168.8 ± 8.3 | 0.62 ± 0.02 | 0.67 ± 0.02 | 139.4 ± 10.2 | 0.68 ± 0.02 | 0.75 ± 0.02 |
| GOLEM-NV-$\ell_1$ | 211.6 ± 4.2 | 0.59 ± 0.02 | 0.22 ± 0.02 | 211.0 ± 4.5 | 0.58 ± 0.03 | 0.22 ± 0.02 |
| NOTEARS | 209.5 ± 1.2 | 0.66 ± 0.02 | 0.15 ± 0.01 | 209.1 ± 1.3 | 0.65 ± 0.02 | 0.15 ± 0.01 |
| PC | 200.5 ± 2.1 | 0.61 ± 0.02 | 0.22 ± 0.01 | 231.0 ± 4.5 | 0.47 ± 0.01 | 0.33 ± 0.01 |
| FGES | 425.6 ± 23.2 | 0.33 ± 0.02 | 0.80 ± 0.01 | 750.0 ± 48.4 | 0.23 ± 0.02 | 1.00 ± 0.00 |
| DAGMA | 253.0 ± 7.5 | 0.49 ± 0.02 | 0.33 ± 0.02 | 252.5 ± 6.4 | 0.49 ± 0.02 | 0.33 ± 0.02 |
| | 100-node ER1 graphs | | | | | |
| | 1000 Samples | | | $10^6$ Samples | | |
| | SHD of CPDAG | Precision of Skeleton | Recall of Skeleton | SHD of CPDAG | Precision of Skeleton | Recall of Skeleton |
| CALM | 26.7 ± 3.6 | 0.90 ± 0.01 | 0.99 ± 0.00 | 17.0 ± 2.9 | 0.96 ± 0.01 | 0.99 ± 0.00 |
| GOLEM-NV-$\ell_1$ | 120.5 ± 6.7 | 0.55 ± 0.02 | 0.75 ± 0.06 | 115.7 ± 4.2 | 0.57 ± 0.02 | 0.68 ± 0.07 |
| NOTEARS | 87.4 ± 2.9 | 0.76 ± 0.01 | 0.74 ± 0.03 | 86.3 ± 2.7 | 0.75 ± 0.01 | 0.78 ± 0.03 |
| PC | 24.7 ± 1.6 | 0.95 ± 0.01 | 0.89 ± 0.01 | 4.2 ± 0.8 | 0.97 ± 0.01 | 1.00 ± 0.00 |
| FGES | 12.1 ± 1.7 | 0.95 ± 0.01 | 0.98 ± 0.00 | 1.0 ± 0.8 | 1.00 ± 0.00 | 1.00 ± 0.00 |
| DAGMA | 152.6 ± 3.9 | 0.55 ± 0.01 | 0.95 ± 0.01 | 150.8 ± 3.3 | 0.55 ± 0.01 | 0.95 ± 0.01 |

Table 5 summarizes the performance comparison between CALM and the baseline methods. The results clearly demonstrate that CALM consistently outperforms NOTEARS, the original GOLEM-NV-$\ell_1$ and DAGMA across all graph structures and sample sizes. This highlights the effectiveness and robustness of incorporating the Gumbel-Softmax approximation to $\ell_0$, moral graph, and hard DAG constraints. It is observed that the results of CALM are not as competitive as those obtained by the PC and FGES methods for sparse graphs such as ER1 graphs. This outcome is expected, given that the continuous optimization in linear likelihood-based formulation struggles with such high levels of nonconvexity.

However, it is worth noting that for ER1 graphs, in the case of 1000 samples, the results of CALM are nearly identical to those of PC. This indicates that in practical scenarios with relatively small sample sizes, even in sparse graphs, CALM is able to compete with the performance of discrete methods like PC. This represents a significant breakthrough.

Furthermore, in more dense graphs, the 50-node ER4 graphs, CALM demonstrates superior performance compared to the PC and FGES methods. This result suggests that in higher-density graphs, CALM enables continuous optimization methods to outperform discrete methods.

The comparison between sparse and dense graphs highlights an important aspect of CALM's performance. While CALM is less competitive than non-differentiable baselines like PC and FGES on sparse graphs, it demonstrates stronger performance on dense graphs. This contrast showcases CALM's ability to handle the increased complexity of dense graph structures.

### 4.7 REVISIONS AND IMPROVEMENTS IN CALM OVER EXISTING METHODS

Our main contributions are to identify and analyze the inconsistency of the $\ell_1$ penalty for learning Markov equivalence classes, and accordingly investigate how to develop a differentiable approach that mitigates this issue, leading to a more practical and robust differentiable approach for learning Markov equivalence classes. As demonstrated in our experiments, our proposed method, CALM, outperforms existing differentiable methods across all settings, including sparse and dense graphs.

In comparison to GOLEM-NV-$\ell_1$, our method introduces several key revisions. First, we address the inconsistency of $\ell_1$-penalties by incorporating a masking approach to approximate the $\ell_0$-penalty. Second, we leverage the moral graph to reduce the search space, which simplifies the optimization process and significantly improves convergence. Additionally, we replace the soft DAG constraints in GOLEM-NV-$\ell_1$ with hard constraints, using a quadratic penalty method inspired by Ng et al. (2022a), leading to a more refined optimization process and ensuring the final graph is always a DAG without requiring post-processing, which often introduces errors in soft constraint methods.

Relative to NOTEARS, our method not only resolves the inconsistency of $\ell_1$-penalties and benefits from the moral graph but is also robust to general non-equal noise variance cases. While NOTEARS assumes equal noise variance, limiting its applicability and causing its performance to degrade after data standardization, our method remains effective across both standardized and non-standardized data.

Overall, these enhancements, including robust parameter tuning, ensure that CALM consistently outperforms other differentiable approaches. It demonstrates superior SHD of CPDAG, skeleton precision, and skeleton recall, across diverse graph types and densities, establishing itself as a more practical and reliable approach for real-world applications.

## 5 CONCLUSION

Our work begins by identifying the inconsistency of $\ell_1$-penalized likelihood in differentiable structure learning for the linear Gaussian case. To address this and improve performance, we propose CALM, which optimizes an $\ell_0$-penalized likelihood with hard acyclicity constraints and incorporates moral graphs. Our results show that CALM, particularly with Gumbel-Softmax $\ell_0$ approximation, significantly outperforms GOLEM-NV-$\ell_1$ and NOTEARS across various graph types and sample sizes. In sparse graphs like ER1, CALM's performance rivals PC with smaller samples, while in dense graphs like ER4, it achieves the best results among all baseline methods. CALM also maintains robust performance both before and after data standardization. Future work includes extending CALM to nonlinear models and integrating advanced optimization techniques for further improvements in linear models.

## REPRODUCIBILITY STATEMENT

Our code will be released publicly upon acceptance of this paper. The implementation details and parameter settings of the CALM-related experiments and baseline methods are mentioned in Section 4 and Appendix A. The details for the experiments proving the inconsistency of $\ell_1$ penalty are provided in Section 3.2.

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

## A  FURTHER IMPLEMENTATION DETAILS FOR SECTION 4

This appendix provides additional implementation details for experiments in Section 4 to ensure clarity and reproducibility of the experiments. We describe our moral graph estimation algorithm; our implementation details of Gumbel-Softmax-based $\ell_0$-penalty and hard DAG constraints in our proposed CALM method and the experiments with hard DAG constraints in Section 4.3; our implementation details of Gumbel-Softmax-based $\ell_0$-penalty and soft DAG constraints in the experiments with soft DAG constraints in section 4.3; our way to achieve infinite samples; our implementation details for STG and Tanh; our implementation details for baseline methods. In some cases in Section 4, particularly when using soft constraints or certain baseline methods, the resulting structures are always not guaranteed to be DAGs after thresholding. To address this, we applied a postprocessing step where edges with the smallest absolute weights were iteratively removed until the structure formed a valid DAG.

### A.1  MORAL GRAPH ESTIMATION

We estimate the moral graph $M$ using the Incremental Association Markov Blanket (IAMB) algorithm (Tsamardinos et al., 2003), which is modified from an implementation available on GitHub: `https://github.com/wt-hu/pyCausalFS/blob/master/pyCausalFS/CBD/MBs/IAMB.py`.

### A.2  IMPLEMENTATION DETAILS OF GUMBEL-SOFTMAX-BASED $\ell_0$-PENALTY AND HARD DAG CONSTRAINTS (CALM AND SECTION 4.3 HARD DAG CONSTRAINT EXPERIMENTS)

In this subsection, we describe the implementation details of the Gumbel-Softmax-based $\ell_0$-penalty and the hard DAG constraints applied in both the CALM method and the experiments that use hard DAG constraints in Section 4.3. We describe four components: (1) Gumbel-Softmax-based $\ell_0$-penalty's initialization, parameter settings and final thresholding; (2) our DAG constraint formulation; and (3) our way to enforce hard DAG constraint and its parameter choices.

**Gumbel-Softmax-based $\ell_0$-penalty**  As introduced in Section 4.1, we use the Gumbel-Softmax mask $g_\tau(U)$ to approximate the $\ell_0$-penalty. The logits matix $U$ is initialized as a zero matrix. The temperature parameter $\tau$ is set to 0.5. The learned parameter matrix $P$ is initialized uniformly between -0.001 and 0.001, following a uniform distribution, to provide a small range of values for the initial weights. This ensures that the learned structure is unbiased at the start of optimization. The hyperparameter $\lambda_1$, which controls the strength of the $\ell_0$-penalty in our method, is set to 0.005.

Additionally, for result thresholding, we follow the approach from Ng et al. (2022b). Specifically, After obtaining the learned logits matrix $U$, we compute $\sigma(U/\tau)$, filter it by the moral graph $M$, and then apply a threshold of 0.5. The resulting matrix is used to compute SHD of CPDAG, precision of skeleton, and recall of skeleton.

**DAG constraint formulation**  The DAG constraint that we use is adapted from $H(B) = \text{Tr}\left(\left(I + \frac{1}{d}B \circ B\right)^d\right) - d$, which proposed by Yu et al. (2019), due to its computational efficiency. In our method, we slightly modified it. Since the mask generated by gumbel softmax and a moral graph, $M \circ g_\tau(U)$, is already non-negative (bounded between 0 and 1), we replace the element-wise multiplication $B \circ B$ with a single mask $M \circ g_\tau(U)$ (for experiments in Section 4.3 where the moral graph is not incorporated, we replace $B \circ B$ with $g_\tau(U)$ alone, which is also bounded between 0 and 1). Hence, the final DAG constraint we used is $H(M \circ g_\tau(U)) = \text{Tr}\left(\left(I + \frac{1}{d}M \circ g_\tau(U)\right)^d\right) - d$ (for experiments in Section 4.3 without the moral graph, this becomes $H(g_\tau(U)) = \text{Tr}\left(\left(I + \frac{1}{d}g_\tau(U)\right)^d\right) - d$). This simplifies the computation while still ensuring the final result $B = M \circ g_\tau(U) \circ P$ is a DAG when $H(M \circ g_\tau(U)) = 0$.

**Enforcing the hard DAG constraint**  Unlike NOTEARS(Zheng et al., 2018), which uses the augmented Lagrangian method (ALM), we use the quadratic penalty method (QPM) (Ng et al., 2022a)

to enforce hard DAG constraints. As noted in Ng et al. (2022a), QPM always yields experimental results consistent with ALM. In our implementation of QPM, $\rho$ serves as a penalty parameter that increases iteratively across optimization steps, and optimization continues until $h(B)$ falls below a predefined threshold. This ensures the final solution satisfies the DAG constraint in most of the cases, progressively tightening the constraint over iterations until convergence at a valid solution. In CALM and the experiments with hard constraints in Section 4.3, the $\rho$ starts at $10^{-5}$ and is gradually increased by a factor of 3 after each subproblem iteration (a block of 40,000 iterations), continuing until the DAG constraint $h$ falls below a threshold of $10^{-8}$. For the optimizer, we choose the Adam optimizer with a learning rate of 0.001.

### A.3 Implementation Details of Gumbel-Softmax-based $\ell_0$-Penalty and Soft DAG Constraints (Section 4.3 Soft DAG Constraint Experiments)

Here, we show implementation and parameters for experiments with soft constraint in Section 4.3. The implementation of the Gumbel-Softmax-based $\ell_0$-penalty and the DAG constraint formulation in the soft DAG constraint experiments follows the same parameter settings and implementation details as outlined in Appendix A.2, including the Gumbel-Softmax mask $g_\tau(U)$, initialization of the logits matrix $U$ and parameter matrix $P$, temperature $\tau$, hyperparameter $\lambda_1$, result thresholding and DAG constraint formulation.

However, as these experiments use soft DAG constraints, we incorporate the DAG constraint as a penalty term in the objective function rather than enforcing it strictly as a hard constraint. In this case, the penalty weight for the DAG constraint, denoted as $\lambda_2$, is set to 0.1. Unlike hard constraints, which are enforced using QPM, the soft constraint is optimized with a single run of 40,000 iterations. We use the Adam optimizer with a learning rate of 0.001.

### A.4 The way to achieve infinity samples

A significant portion of our experiments is conducted under the assumption of infinite sample size. To achieve infinite samples, we use the true covariance matrix, which is calculated as

$$\Sigma^* = (I - B^*)^{-\top} \Omega^* (I - B^*)^{-1}.$$

where $B^*$ is the true adjacency matrix (ground truth DAG) and $\Omega^*$ is the true noise variance matrix. The true covariance matrix $\Sigma^*$ is then substituted into the likelihood term of the objective function, allowing us to simulate the infinite sample case. Specifically, we substitute $\Sigma^*$ for $\Sigma$ in the likelihood term of our method's objective function $\mathcal{L}(M \circ g_\tau(U) \circ P; \Sigma)$. When data standardization is required for infinite samples, we simply compute the standardized covariance matrix, $\Sigma^*_{\text{std}}$ and substitute $\Sigma^*_{\text{std}}$ for $\Sigma$ in $\mathcal{L}(M \circ g_\tau(U) \circ P; \Sigma)$.

### A.5 Implementation details for STG and Tanh

In this section, we provide the implementation details and parameter selection for the STG (Stochastic Gates) and Tanh (Hyperbolic Tangent) methods used to approximate the $\ell_0$ penalty in our experiments in section 4.5 and Appendix B.

The specific details for each method are as follows:

- **STG (Stochastic Gates)** (Yamada et al., 2020): Following the approach of Yamada et al. (2020), we approximate the $\ell_0$ penalty using a stochastic gate mechanism. We define $Z \in \mathbb{R}^{d \times d}$ as the matrix of gates $z_{ij}$, which represents the mask (similar to the role of $g_\tau(U)$ in Section 4.1). $z_{ij}$ is defined as a clipped, mean-shifted, Gaussian random variable $z_{ij} = \max(0, \min(1, \mu_{ij} + \epsilon_{ij}))$, where $\epsilon_{ij} \sim \mathcal{N}(0, \sigma^2)$ and $\sigma$ is fixed during training. $Z$ is then element-wise multiplied by the moral graph $M$. The $\ell_0$ penalty is approximated by the sum of probabilities that the gates are active, which is $\sum_{i,j} \Phi(\mu_{ij}/\sigma) \cdot M_{ij}$, where $\Phi(\cdot)$ is the standard Gaussian CDF, and $M_{ij}$ represents the elements of the moral graph. In our experiments, we initialize $\mu = 0.5$ for all elements, as indicated in the pseudocode in Algorithm 1 of Yamada et al. (2020), and set $\sigma = 0.5$.

- **Tanh (Hyperbolic Tangent)** (Bhattacharya et al., 2021): Following Bhattacharya et al. (2021), the $\ell_0$ penalty is approximated using the hyperbolic tangent function, given by

$\sum_{i,j} \tanh(c|B_{i,j}|)$, where $B_{i,j}$ represents the elements of the weighted adjacency matrix $B$, which has already been restricted by a moral graph. In our experiments, we set the hyperparameter $c$ to 15.

### A.6 Implementation Details for Baseline Methods

In this section, we provide implementation details for the baseline methods used in our experiments, as outlined in Section 4.6 and Appendix C. These include the GOLEM-NV-$\ell_1$, NOTEARS, PC, FGES and DAGMA. As noted by Ng et al. (2024) in their paper's Section 5.1 Observation 2, using a high threshold for edge removal may lead to the wrongful removal of many true edges, causing a significant drop in recall. To mitigate this, we adopted a relatively small threshold of 0.1 in our experiments for the GOLEM-NV-$\ell_1$, NOTEARS and DAGMA. Additionally, for cases where the resulting graph was not a DAG, we applied a post-processing step to remove edges with the smallest absolute values until the resulting graph became a valid DAG.

The specific details for each baseline method are as follows:

- **GOLEM-NV-$\ell_1$** (Ng et al., 2020): We use the parameters recommended by Ng et al. (2020) in their paper. The $\ell_1$ sparsity penalty hyperparameter $\lambda_1$ was set to $2 \times 10^{-3}$, and the soft DAG constraint hyperparameter $\lambda_2$ was set to 5.
- **NOTEARS** (Zheng et al., 2018): We set the $\ell_1$ penalty hyperparameter $\lambda$ to 0.1, and use augmented Lagrangian method to enforce hard DAG constraints like the auther did. All other hyperparameter setting and implementation followed the default setting of the code of Zheng et al. (2018).
- **PC** (Spirtes & Glymour, 1991) and **FGES** (Ramsey et al., 2017): Both PC and FGES were implemented using the `py-causal` package, a Python wrapper of the TETRAD project (Scheines et al., 1998). For PC, we employed the Fisher Z test, and for FGES, we adopted the BIC score (Schwarz, 1978) and set `faithfulnessAssumed = False`.
- **DAGMA** (Bello et al., 2022): We used the code provided by (Bello et al., 2022), setting the `loss_type` to `l2`. The coefficient of the $\ell_1$ penalty, $\lambda_1$, was set to 0.02, following the example provided in their code. All other hyperparameter settings and implementation followed the default settings in the code of Bello et al. (2022).

## B Additional Comparison of $\ell_0$ Approximation Methods for 50-node ER4 Graphs and 100-node ER1 Graphs

This section serves as a supplement to the results presented in Section 4.5, comparing the performance of different $\ell_0$ approximation methods and the original GOLEM-NV-$\ell_1$ on 50-node ER4 graphs and 100-node ER1 graphs. For all methods here, We used infinite samples to eliminate finite sample error. As shown in Table 6, the results here are consistent with the findings in Section 4.5.

Table 6: Comparison of our method using different $\ell_0$ approximations and original GOLEM-NV-$\ell_1$ for 50-node ER4 graphs and 100-node ER1 graphs under both data standardization and no data standardization.

| | **50-node ER4 graphs** | | | | | |
| --- | --- | --- | --- | --- | --- | --- |
| | Standardized | | | Non-Standardized | | |
| | SHD of CPDAG | Precision of Skeleton | Recall of Skeleton | SHD of CPDAG | Precision of Skeleton | Recall of Skeleton |
| CALM | $131.6 \pm 12.3$ | $0.70 \pm 0.02$ | $0.75 \pm 0.02$ | $139.4 \pm 11.3$ | $0.67 \pm 0.02$ | $0.77 \pm 0.02$ |
| CALM-STG | $140.8 \pm 12.3$ | $0.70 \pm 0.02$ | $0.71 \pm 0.03$ | $150.5 \pm 8.5$ | $0.69 \pm 0.02$ | $0.65 \pm 0.02$ |
| CALM-Tanh | $186.2 \pm 4.5$ | $0.69 \pm 0.02$ | $0.41 \pm 0.03$ | $251.6 \pm 9.6$ | $0.46 \pm 0.02$ | $0.45 \pm 0.02$ |
| GOLEM-NV-$\ell_1$ | $212.2 \pm 4.6$ | $0.58 \pm 0.03$ | $0.22 \pm 0.02$ | $294.7 \pm 8.3$ | $0.29 \pm 0.01$ | $0.20 \pm 0.02$ |
| | **100-node ER1 graphs** | | | | | |
| | Standardized | | | Non-Standardized | | |
| | SHD of CPDAG | Precision of Skeleton | Recall of Skeleton | SHD of CPDAG | Precision of Skeleton | Recall of Skeleton |
| CALM | $16.0 \pm 2.9$ | $0.96 \pm 0.01$ | $0.99 \pm 0.00$ | $28.0 \pm 3.6$ | $0.92 \pm 0.01$ | $0.98 \pm 0.01$ |
| CALM-STG | $11.5 \pm 2.2$ | $0.97 \pm 0.01$ | $0.99 \pm 0.00$ | $79.6 \pm 3.5$ | $0.75 \pm 0.01$ | $0.95 \pm 0.01$ |
| CALM-Tanh | $26.2 \pm 3.0$ | $0.93 \pm 0.01$ | $0.97 \pm 0.01$ | $103.4 \pm 3.2$ | $0.67 \pm 0.01$ | $0.88 \pm 0.02$ |
| GOLEM-NV-$\ell_1$ | $109.2 \pm 4.2$ | $0.59 \pm 0.02$ | $0.60 \pm 0.07$ | $217.1 \pm 8.3$ | $0.36 \pm 0.01$ | $0.70 \pm 0.03$ |

We observe that our method using all three different $\ell_0$ approximations—Gumbel-Softmax, STG, and Tanh—consistently outperform GOLEM-NV-$\ell_1$ in both 50-node ER4 graphs and 100-node ER1

graphs. This reinforces the effectiveness of combining $\ell_0$ approximations, moral graph and hard constraints. Among the three $\ell_0$ approximation methods, Gumbel-Softmax achieves the best performance across both graph structures, with strong results observed in both standardized and non-standardized settings. STG shows comparable results to Gumbel-Softmax in the standardized data, but its performance in non-standardized data lags behind that of Gumbel-Softmax, especially in the case of 100-node ER1 graphs.

## C COMPARISON WITH BASELINE METHODS WITHOUT DATA DATA STANDARDIZATION

This appendix serves as a supplement to the results presented in Section 4.6, where we compared the performance of CALM against several baseline methods after data standardization. Here, we provide a comparison of the same methods on 50-node ER1, 50-node ER4, and 100-node ER1 graphs before data standardization, using 1000 and $10^6$ samples. In this section, the moral graph in CALM is estimated from finite samples. Specifically, for the 1000-sample experiments, the moral graph is estimated from 1000 samples, and for the $10^6$-sample experiments, the moral graph is estimated from $10^6$ samples. This ensures consistency in the evaluation across different sample sizes.

It is important to clarify that the results without data standardization are not as significant as those presented in Section 4.6, where data standardization was applied. Nonetheless, we include this comparison in Table 7 for completeness.

Table 7: Comparison with baseline methods for 50-node ER1, 50-node ER4, and 100-node ER1 graphs without data standardization, using 1000 and $10^6$ samples.

| | **50-node ER1 graphs** | | | | | |
|---|---|---|---|---|---|---|
| | 1000 Samples | | | $10^6$ Samples | | |
| | SHD of CPDAG | Precision of Skeleton | Recall of Skeleton | SHD of CPDAG | Precision of Skeleton | Recall of Skeleton |
| CALM | $15.3 \pm 3.4$ | $0.90 \pm 0.02$ | $0.98 \pm 0.00$ | $10.7 \pm 3.4$ | $0.94 \pm 0.02$ | $0.98 \pm 0.01$ |
| GOLEM-NV-$\ell_1$ | $119.4 \pm 5.6$ | $0.35 \pm 0.02$ | $0.76 \pm 0.02$ | $117.3 \pm 6.3$ | $0.36 \pm 0.02$ | $0.75 \pm 0.03$ |
| NOTEARS | $25.7 \pm 2.4$ | $0.74 \pm 0.02$ | $0.98 \pm 0.01$ | $14.7 \pm 2.1$ | $0.86 \pm 0.02$ | $0.98 \pm 0.01$ |
| PC | $11.0 \pm 1.4$ | $0.98 \pm 0.01$ | $0.92 \pm 0.01$ | $2.8 \pm 0.8$ | $0.99 \pm 0.01$ | $0.99 \pm 0.00$ |
| FGES | $8.4 \pm 2.4$ | $0.94 \pm 0.02$ | $0.98 \pm 0.00$ | $1.3 \pm 0.9$ | $1.00 \pm 0.00$ | $1.00 \pm 0.00$ |
| | **50-node ER4 graphs** | | | | | |
| | 1000 Samples | | | $10^6$ Samples | | |
| | SHD of CPDAG | Precision of Skeleton | Recall of Skeleton | SHD of CPDAG | Precision of Skeleton | Recall of Skeleton |
| CALM | $174.6 \pm 7.8$ | $0.60 \pm 0.01$ | $0.71 \pm 0.02$ | $151.4 \pm 10.2$ | $0.65 \pm 0.02$ | $0.75 \pm 0.02$ |
| GOLEM-NV-$\ell_1$ | $293.4 \pm 8.6$ | $0.29 \pm 0.01$ | $0.21 \pm 0.02$ | $291.2 \pm 7.8$ | $0.29 \pm 0.01$ | $0.20 \pm 0.02$ |
| NOTEARS | $191.5 \pm 21.3$ | $0.54 \pm 0.03$ | $0.91 \pm 0.01$ | $178.4 \pm 14.6$ | $0.55 \pm 0.03$ | $0.92 \pm 0.01$ |
| PC | $202.9 \pm 2.0$ | $0.61 \pm 0.02$ | $0.22 \pm 0.01$ | $233.1 \pm 3.4$ | $0.47 \pm 0.01$ | $0.32 \pm 0.01$ |
| FGES | $425.6 \pm 23.2$ | $0.33 \pm 0.02$ | $0.80 \pm 0.01$ | $750.0 \pm 48.3$ | $0.23 \pm 0.02$ | $0.99 \pm 0.00$ |
| | **100-node ER1 graphs** | | | | | |
| | 1000 Samples | | | $10^6$ Samples | | |
| | SHD of CPDAG | Precision of Skeleton | Recall of Skeleton | SHD of CPDAG | Precision of Skeleton | Recall of Skeleton |
| CALM | $37.0 \pm 4.0$ | $0.86 \pm 0.01$ | $0.98 \pm 0.01$ | $29.3 \pm 3.4$ | $0.92 \pm 0.01$ | $0.98 \pm 0.01$ |
| GOLEM-NV-$\ell_1$ | $230.0 \pm 7.9$ | $0.35 \pm 0.01$ | $0.75 \pm 0.02$ | $215.1 \pm 9.4$ | $0.36 \pm 0.02$ | $0.69 \pm 0.04$ |
| NOTEARS | $74.7 \pm 3.7$ | $0.63 \pm 0.01$ | $0.99 \pm 0.01$ | $28.8 \pm 5.2$ | $0.84 \pm 0.03$ | $0.99 \pm 0.00$ |
| PC | $24.5 \pm 1.5$ | $0.95 \pm 0.01$ | $0.89 \pm 0.01$ | $4.2 \pm 0.8$ | $0.97 \pm 0.01$ | $1.00 \pm 0.00$ |
| FGES | $12.1 \pm 1.7$ | $0.95 \pm 0.01$ | $0.98 \pm 0.00$ | $1.0 \pm 0.8$ | $1.00 \pm 0.00$ | $1.00 \pm 0.00$ |

From Table 7, we can find: Firstly, even without data standardization, CALM continues to outperform GOLEM-NV-$\ell_1$ in all cases, demonstrating the robustness of our approach. In particular, the incorporation of the Gumbel-Softmax-based $\ell_0$ approximation, hard DAG constraints, and moral graph still contributes to a substantial improvement over the original GOLEM-NV-$\ell_1$.

Secondly, NOTEARS, which is designed specifically for cases with equal noise variance (EV), performs better before data standardization. This is because, although the noise ratio is set to 16 in the data, the non-equal variance is not as pronounced in the non-standardized data. In contrast, after standardization, the noise ratio becomes more extreme, emphasizing the non-equal variance nature of the data. This explains why NOTEARS performs better in non-standardized settings compared to its performance in standardized settings, even marginally outperforming the CALM in the 100-node ER1 graph with $10^6$ samples. However, this advantage is not meaningful because NOTEARS is inherently based on the EV formulation, which does not align with the non-equal noise variance (NV) setting of our experiments.

Even so, CALM generally surpasses NOTEARS in the non-standardized setting, particularly for the majority of scenarios.

Compared to discrete methods, although the performance gap between CALM and PC slightly widens in the 1000-sample experiments without data standardization, this difference is not significant. One can always standardize the data, and thus, the results from Section 4.6 should be considered more relevant for real-world applications. The pre-standardization results provided here mainly offer insight into the robustness of our method across different settings.

Finally, just as in the results after data standardization in Section 4.6, in more dense graphs, the 50-node ER4 graphs, CALM demonstrates superior performance compared to the PC and FGES methods. This result suggests that in higher-density graphs, CALM enables continuous optimization methods to outperform discrete methods.

# D    COMPARISON OF CALM AND COLiDE SCORE FUNCTIONS

In this section, we compare two different objective functions for linear gaussian non-equal noise variance (NV) formulations: the likelihood-based objective used in GOLEM-NV and CALM and the objective proposed by Saboksayr et al. (2023), named CoLiDE-NV. For a fair comparison, we apply the Gumbel-Softmax approximation for $\ell_0$ to CoLiDE-NV as well, incorporating hard DAG constraints and a moral graph, similar to CALM.

Originally, Saboksayr et al. (2023) propose CoLiDE-NV's score function as

$$\mathcal{S}(B; \Sigma; \Omega) = \frac{1}{2}\text{Tr}\left(\Omega^{-\frac{1}{2}}(I - B)^\top \Sigma (I - B)\right) + \frac{1}{2}\text{Tr}(\Omega^{\frac{1}{2}}) + \lambda_1 \|B\|_1.$$

Unlike the GOLEM-NV model, CoLiDE-NV did not profile out the noise, so the $\Omega$ was kept in the score function. Here, $\Sigma$ is the sample covariance matrix. Also, CoLiDE-NV still used the $\ell_1$ penalty. Since we have shown in Section 3 that $\ell_1$ penalty often leads to inconsistent solutions, we substitute the $\ell_1$ penalty in CoLiDE-NV with the $\ell_0$ penalty, approximated by Gumbel-Softmax. Like CALM, we also incorporate hard DAG constraints and a moral graph to CoLiDE-NV as well. This yields the CoLiDE-$\ell_0$-hard-moral formulation by defining $\mathcal{G}(B; \Sigma; \Omega) = \frac{1}{2}\text{Tr}\left(\Omega^{-\frac{1}{2}}(I - B)^\top \Sigma (I - B)\right) + \frac{1}{2}\text{Tr}(\Omega^{\frac{1}{2}})$,

$$\min_{U \in \mathbb{R}^{d \times d}, P \in \mathbb{R}^{d \times d}} \mathcal{G}(M \circ g_\tau(U) \circ P; \Sigma; \Omega) + \lambda_1 \|M \circ g_\tau(U)\|_1, \quad \text{subject to} \quad h(M \circ g_\tau(U)) = 0.$$

Table 8: Comparison of CALM and CoLiDE-$\ell_0$-hard-moral across 50-node ER1, 50-node ER4, and 100-node ER1 graphs under both No Standardization and Standardization.

| | **50-node ER1 graphs** | | | | | |
| --- | --- | --- | --- | --- | --- | --- |
| | No Standardization | | | Standardization | | |
| | SHD of CPDAG | Precision of Skeleton | Recall of Skeleton | SHD of CPDAG | Precision of Skeleton | Recall of Skeleton |
| CALM | 9.9 ± 3.4 | 0.95 ± 0.02 | 0.99 ± 0.01 | 5.5 ± 1.9 | 0.98 ± 0.01 | 0.99 ± 0.00 |
| CoLiDE-$\ell_0$-hard-moral | 42.5 ± 2.6 | 0.73 ± 0.02 | 0.99 ± 0.00 | 56.2 ± 2.7 | 0.69 ± 0.02 | 0.98 ± 0.01 |
| | **50-node ER4 graphs** | | | | | |
| | No Standardization | | | Standardization | | |
| | SHD of CPDAG | Precision of Skeleton | Recall of Skeleton | SHD of CPDAG | Precision of Skeleton | Recall of Skeleton |
| CALM | 139.4 ± 11.3 | 0.67 ± 0.02 | 0.77 ± 0.02 | 131.6 ± 12.3 | 0.70 ± 0.02 | 0.75 ± 0.02 |
| CoLiDE-$\ell_0$-hard-moral | 157.2 ± 6.2 | 0.62 ± 0.01 | 0.82 ± 0.01 | 185.4 ± 3.4 | 0.67 ± 0.01 | 0.51 ± 0.01 |
| | **100-node ER1 graphs** | | | | | |
| | No Standardization | | | Standardization | | |
| | SHD of CPDAG | Precision of Skeleton | Recall of Skeleton | SHD of CPDAG | Precision of Skeleton | Recall of Skeleton |
| CALM | 28.0 ± 3.6 | 0.92 ± 0.01 | 0.98 ± 0.01 | 16.0 ± 2.9 | 0.96 ± 0.01 | 0.99 ± 0.00 |
| CoLiDE-$\ell_0$-hard-moral | 88.8 ± 4.2 | 0.71 ± 0.01 | 0.99 ± 0.00 | 114.0 ± 3.5 | 0.66 ± 0.01 | 0.97 ± 0.01 |

We also use the quadratic penalty method (QPM) (Ng et al., 2022a) to enforce hard DAG constraints in CoLiDE-$\ell_0$-hard-moral and the results are shown in Table 8. Table 8 summarizes the performance of CALM and CoLiDE-$\ell_0$-hard-moral across linear gaussian model with 50-node ER1, 50-node ER4, and 100-node ER1 graphs under both data standardization and no data standardization. Here, we consider infinite samples. The results show that CALM consistently outperforms CoLiDE-$\ell_0$-hard-moral in all cases.

This demonstrates that CALM's likelihood-based objective is better suited for non-equal noise variance scenarios in the linear Gaussian case. The CoLiDE-$\ell_0$-hard-moral, despite using the correct $\ell_0$ penalty approximation, does not achieve as good results due to its alternative objective CoLiDE-NV.

# E    ADDITIONAL RESULTS ON OTHER GRAPHS

Table 9: Comparison with baseline methods for 20-node ER4, 50-node SF4, 70-node ER4, and 200-node ER4 graphs under data standardization, using 1000 samples.

| 20-node ER4 graphs (1000 Samples with Data Standardization) | | | |
|---|---|---|---|
| | SHD of CPDAG | Precision of Skeleton | Recall of Skeleton |
| CALM | 64.3 ± 3.1 | 0.67 ± 0.03 | 0.64 ± 0.05 |
| GOLEM-NV-$\ell_1$ | 85.7 ± 3.5 | 0.58 ± 0.03 | 0.51 ± 0.06 |
| NOTEARS | 85.3 ± 1.8 | 0.70 ± 0.05 | 0.20 ± 0.01 |
| PC | 81.3 ± 1.5 | 0.65 ± 0.02 | 0.25 ± 0.01 |
| FGES | 114.0 ± 8.5 | 0.48 ± 0.02 | 0.82 ± 0.02 |
| DAGMA | 92.7 ± 3.6 | 0.59 ± 0.04 | 0.38 ± 0.02 |
| 50-node SF4 graphs (1000 Samples with Data Standardization) | | | |
| | SHD of CPDAG | Precision of Skeleton | Recall of Skeleton |
| CALM | 129.3 ± 22.4 | 0.68 ± 0.05 | 0.72 ± 0.03 |
| GOLEM-NV-$\ell_1$ | 176.7 ± 2.1 | 0.80 ± 0.05 | 0.15 ± 0.01 |
| NOTEARS | 189.3 ± 1.1 | 0.77 ± 0.01 | 0.21 ± 0.03 |
| PC | 160.3 ± 4.3 | 0.91 ± 0.03 | 0.26 ± 0.01 |
| FGES | 157.7 ± 24.4 | 0.60 ± 0.04 | 0.74 ± 0.04 |
| DAGMA | 319.3 ± 3.8 | 0.38 ± 0.02 | 0.45 ± 0.02 |
| 70-node ER4 graphs (1000 Samples with Data Standardization) | | | |
| | SHD of CPDAG | Precision of Skeleton | Recall of Skeleton |
| CALM | 180.7 ± 18.3 | 0.69 ± 0.03 | 0.78 ± 0.03 |
| GOLEM-NV-$\ell_1$ | 291.0 ± 3.6 | 0.58 ± 0.03 | 0.19 ± 0.02 |
| NOTEARS | 291.0 ± 2.6 | 0.70 ± 0.04 | 0.15 ± 0.02 |
| PC | 287.0 ± 2.6 | 0.58 ± 0.02 | 0.21 ± 0.01 |
| FGES | 675.0 ± 28.5 | 0.29 ± 0.01 | 0.81 ± 0.03 |
| DAGMA | 338.0 ± 11.0 | 0.50 ± 0.01 | 0.33 ± 0.03 |
| 200-node ER4 graphs (1000 Samples with Data Standardization) | | | |
| | SHD of CPDAG | Precision of Skeleton | Recall of Skeleton |
| CALM | 351.0 ± 67.1 | 0.77 ± 0.04 | 0.86 ± 0.03 |
| GOLEM-NV-$\ell_1$ | 779.0 ± 9.4 | 0.78 ± 0.04 | 0.19 ± 0.04 |
| NOTEARS | 809.7 ± 16.3 | 0.72 ± 0.04 | 0.17 ± 0.01 |
| PC | 780.0 ± 10.6 | 0.62 ± 0.02 | 0.23 ± 0.02 |
| FGES | 1684.7 ± 205.9 | 0.31 ± 0.03 | 0.80 ± 0.02 |
| DAGMA | 921.7 ± 34.8 | 0.51 ± 0.04 | 0.32 ± 0.01 |

We have expanded our experiments to include additional graph sizes and structures, specifically evaluating 20-node ER4, 50-node SF4, 70-node ER4, and 200-node ER4 graphs with 1000 samples under data standardization.

From the results in Table 9, we observe that CALM consistently outperforms other methods, including PC and FGES, on these dense graphs. This demonstrates the robustness and effectiveness of CALM even in challenging dense graph scenarios.

## F  EXPERIMENTS ON EQUAL NOISE VARIANCE CASES

We conducted experiments on 50-node ER1 graphs with 1000 samples, comparing CALM against other baselines under equal noise variances both with and without data standardization. The results are presented in Table 10. From the results, we observe that under data standardization, CALM shows clear advantages over other differentiable methods (NOTEARS, GOLEM-NV-$\ell_1$, and DAGMA), achieving superior SHD of CPDAG, precision of skeleton and recall of skeleton. The performance of CALM is comparable to PC but remains significantly inferior to FGES. However, comparisons with PC and FGES are not the main focus of this paper, as the challenges of non-convexity in differentiable methods make them less competitive in sparse graphs compared to discrete methods.

Before data standardization, NOTEARS and DAGMA perform better than CALM. This is expected, as both algorithms are specifically designed for the equal noise variance case. However, after data standardization, where noise variances become unequal, the performance of NOTEARS and DAGMA drops significantly. As data standardization is a common and practical preprocessing step in real-world applications, the performance after standardization is more relevant. In this context, CALM consistently outperforms other differentiable methods.

Table 10: Performance comparison of CALM and baselines for 50-node ER1 graphs under equal noise variance. Results are shown for standardized and non-standardized data with 1000 samples.

| Standardized Data | | | |
|---|---|---|---|
| Method | SHD of CPDAG | Precision of Skeleton | Recall of Skeleton |
| CALM | 16.3 ± 4.7 | 0.91 ± 0.02 | 0.98 ± 0.01 |
| GOLEM-NV-$\ell_1$ | 68.7 ± 5.9 | 0.51 ± 0.03 | 0.89 ± 0.04 |
| NOTEARS | 40.3 ± 3.6 | 0.79 ± 0.03 | 0.88 ± 0.03 |
| PC | 12.7 ± 2.7 | 0.98 ± 0.00 | 0.93 ± 0.02 |
| FGES | 0.3 ± 0.3 | 0.99 ± 0.01 | 1.00 ± 0.00 |
| DAGMA | 68.3 ± 8.0 | 0.61 ± 0.04 | 0.97 ± 0.01 |
| Non-Standardized Data | | | |
| Method | SHD of CPDAG | Precision of Skeleton | Recall of Skeleton |
| CALM | 17.3 ± 5.1 | 0.90 ± 0.02 | 0.99 ± 0.01 |
| GOLEM-NV-$\ell_1$ | 125.7 ± 11.6 | 0.36 ± 0.03 | 0.86 ± 0.02 |
| NOTEARS | 5.0 ± 4.1 | 0.94 ± 0.05 | 1.00 ± 0.01 |
| PC | 12.7 ± 2.7 | 0.98 ± 0.00 | 0.93 ± 0.02 |
| FGES | 0.3 ± 0.3 | 0.99 ± 0.01 | 1.00 ± 0.00 |
| DAGMA | 0.3 ± 0.3 | 0.99 ± 0.01 | 1.00 ± 0.00 |

## G  REAL-WORD DATA

We conducted experiments on the Sachs dataset (Sachs et al., 2005), which is commonly utilized in probabilistic graphical model research to analyze the expression levels of proteins and phospholipids within human cells. The dataset contains d=11 variables and n=853 samples, with a ground truth of 17 edges. Our method, CALM, achieved an SHD of CPDAG of 12, outperforming GOLEM-NV-$\ell_1$

(SHD of CPDAG: 13) and NOTEARS (SHD of CPDAG: 22). These results demonstrate the strong performance of CALM on real-world data.

## H  TRADE-OFF BETWEEN RUNTIME AND PERFORMANCE IN CALM

There is a trade-off between the $\ell_0$-based methods proposed and existing methods. While CALM does not demonstrate a runtime advantage over other differentiable methods on small and sparse graphs, it may even require less time than some alternatives on larger graphs. For instance, on 50-node SF4 graphs, CALM takes approximately 2500 seconds per run, compared to 20 seconds for NOTEARS and 150 seconds for GOLEM. However, on 200-node ER4 graphs, CALM takes around 4500 seconds per run, whereas NOTEARS takes about 6500 seconds and GOLEM approximately 3000 seconds.

This demonstrates that while CALM's runtime is not the fastest, it scales reasonably well with the graph size, and its performance does not degrade disproportionately as the number of nodes grows. Furthermore, CALM delivers superior results in terms of structural hamming distance (SHD) of CPDAG, skeleton precision, and skeleton recall, particularly for dense and large graphs. These significant performance improvements justify the additional computational cost.

We believe this trade-off between runtime and performance is acceptable, given the substantial gains in accuracy compared to other differentiable methods.

