# OpenReview forum: "Revisiting Differentiable Structure Learning: Inconsistency of $\ell_1$ Penalty and Beyond"
_ICLR.cc/2025/Conference — Submitted to ICLR 2025_

### Official Review · Reviewer_Rxz6 · 2024-10-28

**Soundness:** 3
**Presentation:** 3
**Contribution:** 2
**Rating:** 6
**Confidence:** 4

**Summary:**

The paper studies the inconsistency of DAG learning under $\ell_1$ regularization. A new method, called CALM, is developed that approximates the (nonconvex) $\ell_0$ norm as a regularizer. Experiments on synthetic data suggest that CALM can outperform existing methods and is robust to data standardization.

**Strengths:**

The problem addressed is relevant and timely as most recent DAG learning methods use an $\ell_1$ regularizer, which is well-known to have limitations in other settings. The paper provides compelling evidence that $\ell_0$ regularization can improve on $\ell_1$.

**Weaknesses:**

1. A large body of work exists studying consistency and other properties of $\ell_1$ and $\ell_0$ regularization in linear regression. See, e.g., Zou (2006), Zhao and Yu (2006), and Hastie et al. (2020) and references therein. The present paper should acknowledge this work.
2. The paper highlights the shortcomings of $\ell_1$ regularization but then uses an $\ell_1$ regularizer in CALM's objective function, albeit on the unweighted adjacency matrix. This approximation needs justification as it is not equivalent to using $\ell_0$ regularization. Also, did the authors explore iterative hard thresholding (as commonly used for $\ell_0$ regularized linear regression)?
3. It is not explicitly stated anywhere in the paper, but the problem with $\ell_1$ is that it (continuously) shrinks the weighted adjacency matrix to zero, setting some elements exactly to zero. In constrast, $\ell_0$ does not impart any shrinkage, it just selects. It is worth making this distinction clear to the reader early on.
4. I am confused regarding the robustness of CALM to data standardization. Is the robustness due to using $\ell_0$ regularization instead of $\ell_1$ regularization? My undertanding was that data standarization issues arise due to the continuous acyclicity characterization, not the form of sparsity penalty.
5. Related to the above, the fact that the error variances are on different scales may determine the impact of the data standardization. Specifically, if the error variances are equal across nodes, then the node variances give insight into the true topological ordering. If they are unequal, the node variances offer much less information, and maybe the algorithm just benefits from having all the gradients on the same scale.
6. How does CALM perform when the true noise variances are equal? I agree unequal variances are more realistic, but the equal variances case (which is the de facto standard in the literature) is also worth studying.
7. DAGMA is mentioned in the paper but not included in the experiments. It should be added as a baseline.

References:
1. Zou, H. (2006). "The adaptive lasso and its oracle properties". Journal of the American Statistical Association 101 (476), pp. 1418–1429.
2. Zhao, P. and Yu, B (2006). "On model selection consistency of lasso". Journal of Machine Learning Research 7, pp. 2541–2563.
3. Hastie T., Tibshirani, R., and Tibshirani, R. (2020). "Best subset, forward stepwise or lasso? Analysis and recommendations based on extensive comparisons". Statistical Science 35 (4), pp. 579–592.

**Questions:**

1. Page 1 Line 34: "PEARL" should not be uppercase.
2. Page 1 Line 37: "economy" should be "economics".
3. Page 4 Line 166: I do not understand the claim that the $\ell_1$ penalty "doesn't guarantee true sparsity". Do you mean it does not guarantee a sparsity level that matches that of the true graph?
4. Table 1: Please add standard errors to the table. Also, the quotation marks in the caption are facing the wrong way (also in other parts of the paper).
5. Page 6 Line 274: "utilizes" should be "utilizing".
6. Page 6 Line 323: There is an error with the Erdos-Renyi citation.
7. Page 7 Line 368: I do not think it is true that hard DAG constraints (implemented by QPM) ensure the final matrix is always a DAG. Only when $\rho\to\infty$ is the graph guaranteed to be a DAG.
8. Numbers are not formatted consistently, e.g., "1000" on Page 4 Line 215 vs. "1,000" on Page 5 Line 217

---

> ### Author Response · Authors · 2024-11-23
> **Responses to Reviewer Rxz6 (part 1/4)**
>
> Thank you very much for your valuable comments and insightful suggestions. We have carefully considered your feedback and addressed each point below.
>
> **Q1**: “A large body of work exists studying consistency and other properties of $\ell_1$ and $\ell_0$ regularization in linear regression. See, e.g., Zou (2006), Zhao and Yu (2006), and Hastie et al. (2020) and references therein. The present paper should acknowledge this work.”
>
> **A1**: Thanks for suggesting these references. We will cite them in the revision.
>
>
> **Q2**：“The paper highlights the shortcomings of $\ell_1$ regularization but then uses an $\ell_1$ regularizer in CALM's objective function, albeit on the unweighted adjacency matrix. This approximation needs justification as it is not equivalent to using $\ell_0$ regularization. Also, did the authors explore iterative hard thresholding (as commonly used for $\ell_0$ regularized linear regression)?”
>
> **A2**: We appreciate this thoughtful comment. We agree that using $\ell_1$ regularizer on the unweighted adjacency matrix is an approximation to $\ell_0$ penalty. This is because directly solving the problem with the $\ell_0$ penalty is NP-hard, and thus we resort to an approximation. The justification of such an approximation is that the lower the temperature of Gumbel softmax, the closer it is to being a binary matrix (and thus closer to being a $\ell_0$ penalty). Furthermore, using the parameterization with Gumbel Softmax allows for a fully differentiable estimation method. We will include this discussion in the revision.
>
> On the other hand, as the reviewer pointed out, iterative hard thresholding is a common approach used for $\ell_0$ regularized linear regression. At the same time, our objective function is considerably more complicated than linear regression, and thus iterative hard thresholding may not perform well here. In light of your suggestion, we will include a comparison with iterative hard thresholding in Section 4.5 of the revision.
>
> **Q3**：“It is not explicitly stated anywhere in the paper, but the problem with $\ell_1$ is that it (continuously) shrinks the weighted adjacency matrix to zero, setting some elements exactly to zero. In constrast, $\ell_0$ does not impart any shrinkage, it just selects.”
>
> **A3**：Great point. We will include this explanation early on in Section 3 of the revision to make this clear.
>
>
> **Q4**: “I am confused regarding the robustness of CALM to data standardization. Is the robustness due to using $\ell_0$ regularization instead of $\ell_1$ regularization? My understanding was that data standardization issues arise due to the continuous acyclicity characterization, not the form of sparsity penalty.“
>
> **A4**: Thanks for this thoughtful question. CALM is relatively robust to data standardization because (1) it does not assume equal noise variances, and (2) the estimation procedure is consistent in the sense that the global minimum corresponds to the true Markov equivalence class (under some conditions) with $\ell_0$ regularization. For other methods, GOLEM with $\ell_1$ regularization is not consistent both before and after data standardization, and thus performs poorly. For NOTEARS, as discussed by Ng et al. (2024), it is not robust to data standardization because the assumption of equal noise variances is violated after data standardization. We will include this discussion in the revision.
>
>
> **Q5**: “Related to the above, the fact that the error variances are on different scales may determine the impact of the data standardization. Specifically, if the error variances are equal across nodes, then the node variances give insight into the true topological ordering. If they are unequal, the node variances offer much less information, and maybe the algorithm just benefits from having all the gradients on the same scale.“
>
> **A5**: Thanks for this insightful comment. Such a study is interesting and provides insight into why certain methods perform well when the noise variances are equal; a similar study has been conducted by Reisach et al. (2021). At the same time, since our method is rather robust to data standardization (i.e., can perform well both before and after standardization), it seems to indicate that our method does not exploit such information that gives insight into the true topological ordering, which can be viewed as an advantage of our method. We will include this discussion in the revision.

---

> ### Author Response · Authors · 2024-11-23
> **Responses to Reviewer Rxz6 (part 2/4)**
>
> **Q6**：“How does CALM perform when the true noise variances are equal? I agree unequal variances are more realistic, but the equal variances case (which is the de facto standard in the literature) is also worth studying.”
>
> **A6**: Thank you for raising this excellent point. We have conducted preliminary experiments on 50-node ER1 graphs with 1000 samples, comparing CALM against other baselines under equal noise variances both with and without data standardization. The results are presented below.  We have updated the manuscript to include these results in Appendix F. We will include the final complete results in the revision.
>
> ***
>
> ### 50-node ER1 graphs (1000 Samples with Data Standardization in Equal Noise Variance Case)
>
> | Method         | SHD of CPDAG       | Precision of Skeleton | Recall of Skeleton |
> |----------------|--------------------|------------------------|--------------------|
> | CALM           | 16.3 ± 4.7        | 0.91 ± 0.02           | 0.98 ± 0.01       |
> | GOLEM-NV-$\ell_1$ | 68.7 ± 5.9        | 0.51 ± 0.03           | 0.89 ± 0.04       |
> | NOTEARS        | 40.3 ± 3.6        | 0.79 ± 0.03           | 0.88 ± 0.03       |
> | PC             | 12.7 ± 2.7        | 0.98 ± 0.00           | 0.93 ± 0.02       |
> | FGES           | 0.3 ± 0.3         | 0.99 ± 0.01           | 1.00 ± 0.00       |
> | DAGMA          | 68.3 ± 8.0        | 0.61 ± 0.04           | 0.97 ± 0.01       |
>
> ***
>
> ### 50-node ER1 graphs (1000 Samples without Data Standardization in Equal Noise Variance Case)
>
> | Method         | SHD of CPDAG       | Precision of Skeleton | Recall of Skeleton |
> |----------------|--------------------|------------------------|--------------------|
> | CALM           | 17.3 ± 5.1        | 0.90 ± 0.02           | 0.99 ± 0.01       |
> | GOLEM-NV-$\ell_1$ | 125.7 ± 11.6      | 0.36 ± 0.03           | 0.86 ± 0.02       |
> | NOTEARS        | 5.0 ± 4.1         | 0.94 ± 0.05           | 1.00 ± 0.01       |
> | PC             | 12.7 ± 2.7        | 0.98 ± 0.00           | 0.93 ± 0.02       |
> | FGES           | 0.3 ± 0.3         | 0.99 ± 0.01           | 1.00 ± 0.00       |
> | DAGMA          | 0.3 ± 0.3         | 0.99 ± 0.01           | 1.00 ± 0.00       |
>
> ***
>
>
> From the results, we observe that under data standardization, CALM shows clear advantages over other differentiable methods (NOTEARS, GOLEM-NV-$\ell_1$​, and DAGMA), achieving superior SHD and precision while maintaining high recall. The performance of CALM is comparable to PC but remains significantly inferior to FGES. However, as mentioned in a previous response, comparisons with PC and FGES are not the main focus of this paper, as the challenges of non-convexity in differentiable methods make them less competitive in sparse graphs compared to discrete methods.
> Before data standardization, NOTEARS and DAGMA perform better than CALM. This is expected, as both algorithms are specifically designed for the equal noise variance case. However, after data standardization, where noise variances become unequal, the performance of NOTEARS and DAGMA drops significantly. As data standardization is a common and practical preprocessing step in real-world applications, the performance after standardization is more relevant. In this context, CALM consistently outperforms other differentiable methods.

---

> ### Author Response · Authors · 2024-11-23
> **Responses to Reviewer Rxz6 (part 3/4)**
>
> **Q7**： “DAGMA is mentioned in the paper but not included in the experiments. It should be added as a baseline.”
>
> **A7**： Thank you for your insightful suggestion.  We have added DAGMA as a baseline for 50-node ER1, 50-node ER4, and 100-node ER1 graphs under both 1000 and $10^6$ samples. The experimental settings, including running the experiments on the same 10 seeds, are consistent with those described in Table 5 of the initial submission. The results are summarized below:
> ***
>
> ### 50-node ER1 graphs
>
> | Method         | SHD of CPDAG (1000 Samples) | Precision of Skeleton (1000 Samples) | Recall of Skeleton (1000 Samples) | SHD of CPDAG ($10^6$ Samples) | Precision of Skeleton ($10^6$ Samples) | Recall of Skeleton ($10^6$ Samples) |
> |----------------|-----------------------------|---------------------------------------|-----------------------------------|--------------------------------|----------------------------------------|----------------------------------------|
> | CALM           | 12.1 ± 2.7                 | 0.93 ± 0.01                          | 0.98 ± 0.00                     | 7.0 ± 2.5                     | 0.97 ± 0.01                           | 0.98 ± 0.01                           |
> | DAGMA          | 73.3 ± 4.0                 | 0.57 ± 0.02                          | 0.95 ± 0.01                     | 70.6 ± 3.4                    | 0.59 ± 0.02                           | 0.95 ± 0.01                           |
>
> ***
>
> ### 50-node ER4 graphs
>
> | Method         | SHD of CPDAG (1000 Samples) | Precision of Skeleton (1000 Samples) | Recall of Skeleton (1000 Samples) | SHD of CPDAG ($10^6$ Samples) | Precision of Skeleton ($10^6$ Samples) | Recall of Skeleton ($10^6$ Samples) |
> |----------------|-----------------------------|---------------------------------------|-----------------------------------|--------------------------------|----------------------------------------|----------------------------------------|
> | CALM           | 168.8 ± 8.3                | 0.62 ± 0.02                          | 0.67 ± 0.02                     | 139.4 ± 10.2                  | 0.68 ± 0.02                           | 0.75 ± 0.02                           |
> | DAGMA          | 253.0 ± 7.5                | 0.49 ± 0.02                          | 0.33 ± 0.02                     | 252.5 ± 6.4                   | 0.49 ± 0.02                           | 0.33 ± 0.02                           |
>
> ***
>
> ### 100-node ER1 graphs
>
> | Method         | SHD of CPDAG (1000 Samples) | Precision of Skeleton (1000 Samples) | Recall of Skeleton (1000 Samples) | SHD of CPDAG ($10^6$ Samples) | Precision of Skeleton ($10^6$ Samples) | Recall of Skeleton ($10^6$ Samples) |
> |----------------|-----------------------------|---------------------------------------|-----------------------------------|--------------------------------|----------------------------------------|----------------------------------------|
> | CALM           | 26.7 ± 3.6                 | 0.90 ± 0.01                          | 0.99 ± 0.00                     | 17.0 ± 2.9                     | 0.96 ± 0.01                           | 0.99 ± 0.00                           |
> | DAGMA          | 152.6 ± 3.9                | 0.55 ± 0.01                          | 0.95 ± 0.01                     | 150.8 ± 3.3                    | 0.55 ± 0.01                           | 0.95 ± 0.01                           |
>
> ***
>
>
>
> It can be seen from the results that CALM demonstrates significant advantages over DAGMA across all scenarios. We have updated the manuscript’s Section 4.6 and Table 5 to include DAGMA's results. Furthermore, we have updated Appendix A.6 in the manuscript to include DAGMA's implementation details.
>
> **Q8**：“Page 4 Line 166: I do not understand the claim that the $\ell_1$​ penalty “doesn't guarantee true sparsity”. Do you mean it does not guarantee a sparsity level that matches that of the true graph?”
>
> **A8**: Thanks for this question. As you nicely mentioned, by “true sparsity”, we intend to convey that the sparsity level (i.e., number of edges) does not match that of the true graph. We will clarify this in the revision to avoid any possible confusion.

---

> ### Author Response · Authors · 2024-11-23
> **Responses to Reviewer Rxz6 (part 4/4)**
>
> **Q9**：“Table 1: Please add standard errors to the table. Also, the quotation marks in the caption are facing the wrong way (also in other parts of the paper).”
>
> **A9**：Thank you for your valuable suggestion. We have updated the manuscript to include the standard errors to Table 1, as shown below:
>
> ***
>
> | Metric                                  | $B^*$         | $B_{\ell_1}$   | Proportion of $\tilde{B}$ with $\|\|\tilde{B}\|\|_1 < \|\|B^*\|\|_1$ |
> |-----------------------------------------|---------------|----------------|-----------------------------------------------------------------------|
> | Average $\ell_1$ norm                  | 10.04 ± 0.04  | 4.22 ± 0.03    | 77.86% ± 0.46%                                                      |
> | Average $\ell_0$ norm (Number of Edges)| 8.0 ± 0.0     | 22.74 ± 0.15   |                                                                       |
> | Average SHD of CPDAG                   | 0 ± 0.0       | 19.97 ± 0.17   |                                                                       |
>
> ***
>
> Also, we have updated the manuscript to correct the issue of quotation marks facing the wrong way in the captions and other parts of the paper.
>
> **Q10**: “Page 7 Line 368: I do not think it is true that hard DAG constraints (implemented by QPM) ensure the final matrix is always a DAG. Only when $\rho\rightarrow\infty$ is the graph guaranteed to be a DAG.”
>
> **A10**：We appreciate this insightful question. We agree with the reviewer that only when $\rho\rightarrow\infty$ the graph is guaranteed to be a DAG, according to Ng et al. (2022). We will update the sentence in Line 368 to “hard constraints ensure that the final graph is very close to a DAG, which thus makes it easier to transform the estimated solution into the desired structure” in the revision.
>
> **Q11**: Minor issues such as typos and formatting issue
>
> **A11**: Thanks for your careful reading. We will fix these typos and formatting issues in the final version, and carefully proofread the paper.

---

> ### Comment · Reviewer_Rxz6 · 2024-11-25
>
> I thank the authors for their response. My concerns are mainly addressed, so I have revised my score from 5 to 6.

---

> > ### Author Response · Authors · 2024-11-30
> > **Acknowledgment for Reviewer Rxz6’s Feedback and Suggestions**
> >
> > Dear Reviewer Rxz6,
> >
> > We sincerely thank the reviewer for your constructive feedback, the time you have dedicated, and your recognition of our work. We will incorporate all the suggestions in the revised manuscript.
> >
> > Best regards,
> >
> > Authors of Paper 4754

---

### Official Review · Reviewer_1ULx · 2024-11-02

**Soundness:** 2
**Presentation:** 2
**Contribution:** 2
**Rating:** 6
**Confidence:** 3

**Summary:**

The article studies differentiable structure learning methods for learning directed acyclic graphs (DAG) related to probabilistic graphical models. Particularly, the continuous optimization approach  with  $\ell_1$ norm penalty (to learn sparse DAG) is studied and experimentally it is shown that this approach is inconsistent compared to the solutions obtained using   $\ell_0$ norm penalty. Next, a new hybrid differentiable structure learning method is proposed based on $\ell_0$ norm likelihood and hard acyclicity constraints. Experimental results illustrate the performance of the proposed method.

**Strengths:**

Strengths:
1. Certain inconsistencies related to the $\ell_1$ penalty based structure learning is investigated.
2. New method is proposed for  structure learning based on   $\ell_0$ norm penalty.
3. Numerical results on few graphs show improved results.

**Weaknesses:**

Weakness:
1. Many aspects of the study are unclear.
2. Significance of the study and results are unclear, and novelty seems limited.
3. Comprehensive numerical study is lacking.

**Questions:**

I have the following comments about the paper:

1. Many aspects of the presented study can be improved, in order to make the contributions of the paper clear.

i. Firstly,  the term "inconsistency" seems to be used without formal definition. It would be helpful if the authors can provide a formal definition of "inconsistency" as used in this context.

ii. It is not clear whether the $\ell_1$ norm penalty based approach is a popular approach for structure learning.  The authors could provide context on the prevalence and importance of $\ell_1$  norm penalty approaches in structure learning, particularly since in the experiment results presented, it appears the $\ell_1$ penalty approaches seem to perform poorly compared to other baselines such as FGES or PC.

iii.  It is claimed that the $\ell_1$ penalty approach is particularly bad for Linear Gaussian case, and Markov equivalence classes. However, it is not clear why are Markov equivalence classes important, and where do these appear in structure learning applications?  Authors can include a brief explanation of why Markov equivalence classes are significant in structure learning, and provide examples of real-world applications where linear Gaussian assumptions are commonly used.

iv. In order to establish the inconsistencies, the $\ell_1$ approach is compared to an  $\ell_0$ norm penalty approach, by solving the eqn in line 154. However, details on how this problem is solved is missing. Authors to provide more details on the method used to solve the ℓ $\ell_0$ norm penalty problem, including any computational challenges or trade-offs compared to the $\ell_1$ approach.

v. Motivation for using structural Hamming distance (SHD) and how it is computed are not discussed.

2. Given the above points, the significance of the presented results are not obvious to readers. Moreover, the proposed new approach CALM (Continuous and Acyclicity-constrained L0-penalized likelihood with estimatedMoral graph), contains two components, the Gumbel Softmax based $\ell_0$  penalty  approximation, and incorporating the moral graph and hard DAG constraints. But, both these methods seem have been proposed in previous papers, and it is not clear what is the novelty in the proposed method, compared to the previous works such as Ng et al. (2022b); Kalainathan et al. (2022), Nazaret et al. (2024), and Ng et al. (2024).

Authors could clearly articulate the novel aspects of CALM compared to previous work, highlighting any unique combinations or modifications of existing techniques. Also, they can provide a more detailed description of how the optimization problem is solved, including computational complexity and any approximations used.

3. Numerical study is presented on three relatively small synthetic graphs. It is not clear if the CALM method is compared to all structure learning baselines.  Authors can expand their experiments to include a wider range of graph sizes and structures, and provide a more comprehensive comparison with other state-of-the-art structure learning methods.
Also, the performance of CALM approaches seems similar to the FGES, and in some cases they are worse than FGES. So, it is not clear what are the advantages of the proposed method. Authors could include runtime comparisons to better illustrate the practical advantages of their method.

4. Minor Comment:
i.  Why are uppercase letters used for both matrices and vectors?
ii. In line 97, X is nxd, but B is dxd, how do we compute B^TX?
iii. How is SHD computed?
iv. What is PC method? What does PC stand for?

---

> ### Author Response · Authors · 2024-11-23
> **Responses to Reviewer 1ULx (part 1/4)**
>
> Thank you very much for your valuable comments and insightful suggestions. We have carefully considered your feedback and addressed each point below.
>
> **Q1.1** : “Firstly, the term “inconsistency” seems to be used without formal definition.”
>
> **A1.1** : Thanks for asking this question, which helps improve the clarity of our paper. By inconsistency, we intend to convey that, in the large sample limit, the algorithm (or estimator) cannot return a DAG that is Markov equivalent to the ground truth. We will include this definition in the revision to make it clear.
>
> **Q1.2** : “It is not clear whether the $\ell_1$ norm penalty based approach is a popular approach for structure learning.”
>
> **A1.2** : We appreciate this insightful question. Continuous optimization has received considerable attention in causal structure learning (e.g., see [1] for a review), of which a key ingredient is $\ell_1$ penalty to enforce sparsity. The $\ell_1$ penalty has been used in a wide range of settings [1], including those where the structure is fully identifiable (Zheng et al. 2018 & 2020), and those where the structure is only identifiable up to Markov equivalence class (Ng et al. 2020). In this work, we focus on the latter case; specifically, we demonstrate that $\ell_1$ penalty is inconsistent and propose alternative algorithms to remedy such issues. Such a study also highlights our contribution in formally studying such inconsistency issues given the popularity of $\ell_1$ penalty.
>
> **Q1.3** : “Authors can include a brief explanation of why Markov equivalence classes are significant in structure learning, and provide examples of real-world applications where linear Gaussian assumptions are commonly used.”
>
> **A1.3** : Thanks for this comment. Markov equivalence classes play an important role in structure learning, because without any parametric assumption, one can only identify the Markov equivalence class of the ground truth DAG, e.g., by making use of conditional independence test (Spirtes et al. 2002). Therefore, developing methods to learn Markov equivalence classes have been a major research area in structure learning [2]. Regarding real-world applications, linear Gaussian assumptions may be used to model psychological data [3] and fMRI data.
>
> **Q1.4** : “Authors to provide more details on the method used to solve the $\ell_0$ norm penalty problem, including any computational challenges or trade-offs compared to the $\ell_1$ approach.”
>
> **A1.4** : The optimization problem involving $\ell_0$ norm in Line 154 is computationally challenging as the search space of possible DAGs grows super-exponentially in the number of variables. Exact search procedures can be used, such as those based on dynamic programming [4] or sparsest permutation (Raskutti & Uhler, 2018). (In this work, we adopt a procedure based on sparsest permutation.) Such procedures typically do not scale up to a large number of variables, and thus one may resort to greedy search procedures such as GES. On the other hand, the $\ell_1$ approach allows for continuous optimization, which is often more efficient or scalable.
>
> **Q1.5** : “Motivation for using structural Hamming distance (SHD) and how it is computed are not discussed.”
>
> **A1.5** : Structural Hamming distance is a common metric used in structure learning. Specifically, it calculates the number of edge operations (including addition, deletion, and reversal) needed to transform one structure into another. We will include this definition in the revision.

---

> ### Author Response · Authors · 2024-11-23
> **Responses to Reviewer 1ULx (part 2/4)**
>
> **Q2** : “Given the above points, the significance of the presented results are not obvious to readers…both these methods seem have been proposed in previous papers, and it is not clear what is the novelty in the proposed method…Authors could clearly articulate the novel aspects of CALM compared to previous work, highlighting any unique combinations or modifications of existing techniques. Also, they can provide a more detailed description of how the optimization problem is solved, including computational complexity and any approximations used.”
>
> **A2** : Thanks for this comment. In our paper, we focus on identifying and analyzing the inconsistency of the $\ell_1$ penalty for learning Markov equivalence classes, and accordingly investigate how to develop a differentiable approach that mitigates this issue. This issue has not been thoroughly investigated in prior work. Our primary goal is not to propose a completely novel  method but to develop a more practical and robust differentiable approach for learning Markov equivalence classes. This work aims to improve performance and robustness, enabling broader applicability to real-world problems.
>
> In summary, we make the following modifications to improve performance and robustness and make it practically more useful. (1) Masking Approach:  Our method leverages the masking approach in Ng et al. (2022b); Kalainathan et al. (2022) for learning equivalence classes in linear Gaussian case. (2) Moral Graph: To reduce the search space, we leverage the moral graph, which significantly simplifies the optimization process and improves convergence.   (3) Parameter Tuning: We carefully tune the hyperparameters to ensure optimal performance. For example, we determined the hyperparameter $\lambda_1$, which controls the $\ell_0$-penalty, through extensive experiments. Various values such as 0.0005, 0.05, and 0.5 were tested, with 0.005 consistently yielding the best results across different settings. Other parameters were also carefully tuned to select the optimal ones, with the final choices detailed in Appendix A of the paper. (4) Hard DAG Constraints: We utilize hard DAG constraints. To solve the optimization problem effectively, we employ a quadratic penalty method inspired by Ng et al. (2022a).
>
> In comparison to GOLEM-NV-$\ell_1$​, our method introduces several key revisions. First, we address the inconsistency of $\ell_1$​-penalties by incorporating a masking approach to approximate the $\ell_0$-penalty. Second, we leverage the moral graph to reduce the search space, which simplifies the optimization process and significantly improves convergence (see Section 4.3 for an analysis of the impact of the moral graph). Additionally, we replace the soft DAG constraints in GOLEM-NV-$\ell_1$ with hard constraints, using a quadratic penalty method inspired by Ng et al. (2022a), leading to a more refined optimization process and ensuring the final graph is always a DAG without requiring post-processing, which often introduces errors in soft constraint methods (see Section 4.3 for an analysis of the impact of soft/hard DAG constraints).
> Relative to NOTEARS, our method not only resolves the inconsistency of $\ell_1$​-penalties and benefits from the moral graph but is also robust to general non-equal noise variance cases. While NOTEARS assumes equal noise variance, limiting its applicability and causing its performance to degrade after data standardization (see Appendix C for an analysis of NOTEARS' results), our method remains effective across both standardized and non-standardized data.
> Overall, these enhancements, including robust parameter tuning, ensure that CALM consistently outperforms other differentiable approaches. It demonstrates superior SHD of CPDAG, skeleton precision, and skeleton recall, across diverse graph types and densities, establishing itself as a more practical and reliable approach for real-world applications.  We have also updated the manuscript to include these revisions and improvements in CALM over existing differentiable methods in Section 4.7.
>
> Regarding the optimization problem, we solve it using  a quadratic penalty method inspired by Ng et al. (2022a), where each subproblem is tackled via gradient-based optimization using the Adam optimizer. The computational complexity is $O(d^3)$ per iteration, which is similar to most other differentiable structure learning methods in the  linear case, such as NOTEARS and GOLEM.
> We have updated the manuscript to include this discussion of the optimization problem in Section 4.2.
>
> We will include a more detailed discussion of these aspects in the revision, clearly articulating the significance of our study and the unique contributions of CALM. We hope this addresses your concerns and provides clarity on the novelty and practical relevance of our work.

---

> ### Author Response · Authors · 2024-11-23
> **Responses to Reviewer 1ULx (part 3/4)**
>
> **Q3** : “Authors can expand their experiments to include a wider range of graph sizes and structures, and provide a more comprehensive comparison with other state-of-the-art structure learning methods.”
>
>
> **A3** : Thank you for pointing out the need for a more comprehensive numerical study. To address this concern, we have expanded our experiments by conducting preliminary evaluations on additional graph sizes and structures, specifically 20-node ER4, 50-node SF4, 70-node ER4, and 200-node ER4 graphs with 1000 samples under data standardization. We have updated the manuscript to include these results in Appendix E and will include the final complete results in the revision. In these additional experiments, we also added DAGMA as a baseline. The results are summarized below:
> ***
>
> ### 20-node ER4 graphs (1000 Samples with Data Standardization)
>
> | Method         | SHD of CPDAG       | Precision of Skeleton | Recall of Skeleton |
> |----------------|--------------------|------------------------|--------------------|
> | CALM           | 64.3 ± 3.1        | 0.67 ± 0.03           | 0.64 ± 0.05       |
> | GOLEM-NV-$\ell_1$ | 85.7 ± 3.5        | 0.58 ± 0.03           | 0.51 ± 0.06       |
> | NOTEARS        | 85.3 ± 1.8        | 0.70 ± 0.05           | 0.20 ± 0.01       |
> | PC             | 81.3 ± 1.5        | 0.65 ± 0.02           | 0.25 ± 0.01       |
> | FGES           | 114.0 ± 8.5       | 0.48 ± 0.02           | 0.82 ± 0.02       |
> | DAGMA          | 92.7 ± 3.6        | 0.59 ± 0.04           | 0.38 ± 0.02       |
>
> ***
>
> ### 50-node SF4 graphs (1000 Samples with Data Standardization)
>
> | Method         | SHD of CPDAG       | Precision of Skeleton | Recall of Skeleton |
> |----------------|--------------------|------------------------|--------------------|
> | CALM           | 129.3 ± 22.4      | 0.68 ± 0.05           | 0.72 ± 0.03       |
> | GOLEM-NV-$\ell_1$ | 176.7 ± 2.1       | 0.80 ± 0.05           | 0.15 ± 0.03       |
> | NOTEARS        | 189.3 ± 1.1       | 0.77 ± 0.01           | 0.21 ± 0.03       |
> | PC             | 160.3 ± 4.3       | 0.91 ± 0.03           | 0.26 ± 0.01       |
> | FGES           | 157.7 ± 4.4       | 0.74 ± 0.04           | 0.74 ± 0.04       |
> | DAGMA          | 319.3 ± 3.8       | 0.38 ± 0.02           | 0.45 ± 0.02       |
>
> ***
>
> ### 70-node ER4 graphs (1000 Samples with Data Standardization)
>
> | Method         | SHD of CPDAG       | Precision of Skeleton | Recall of Skeleton |
> |----------------|--------------------|------------------------|--------------------|
> | CALM           | 180.7 ± 18.3      | 0.69 ± 0.03           | 0.78 ± 0.03       |
> | GOLEM-NV-$\ell_1$ | 291.0 ± 3.6       | 0.58 ± 0.03           | 0.19 ± 0.02       |
> | NOTEARS        | 291.0 ± 2.6       | 0.70 ± 0.04           | 0.15 ± 0.02       |
> | PC             | 287.0 ± 2.6       | 0.58 ± 0.02           | 0.21 ± 0.01       |
> | FGES           | 675.0 ± 28.5      | 0.29 ± 0.01           | 0.81 ± 0.03       |
> | DAGMA          | 338.0 ± 11.0      | 0.50 ± 0.01           | 0.33 ± 0.03       |
>
>
> ***
>
> ### 200-node ER4 graphs (1000 Samples with Data Standardization)
>
> | Method         | SHD of CPDAG       | Precision of Skeleton | Recall of Skeleton |
> |----------------|--------------------|------------------------|--------------------|
> | CALM           | 351.0 ± 67.1      | 0.77 ± 0.04           | 0.86 ± 0.03       |
> | GOLEM-NV-$\ell_1$ | 779.0 ± 9.4       | 0.78 ± 0.04           | 0.19 ± 0.04       |
> | NOTEARS        | 809.7 ± 16.3      | 0.72 ± 0.04           | 0.17 ± 0.01       |
> | PC             | 780.0 ± 10.6      | 0.60 ± 0.04           | 0.23 ± 0.02       |
> | FGES           | 1684.7 ± 205.9    | 0.31 ± 0.04           | 0.81 ± 0.02       |
> | DAGMA          | 921.7 ± 34.8      | 0.51 ± 0.04           | 0.32 ± 0.01       |
>
> ***
>
> From the results above, we observe that CALM consistently outperforms other methods, including PC and FGES, on these dense graphs. This demonstrates the robustness and effectiveness of CALM even in challenging dense graph scenarios.

---

> ### Author Response · Authors · 2024-11-23
> **Responses to Reviewer 1ULx (part 4/4)**
>
> **Q4** : “Also, the performance of CALM approaches seems similar to the FGES, and in some cases they are worse than FGES. So, it is not clear what are the advantages of the proposed method.”
>
> **A4** : Thanks for your question. We would like to clarify that our primary focus is on improving differentiable structure learning methods for learning Markov equivalence classes, as these methods have recently gained significant attention in the research community. Therefore, our main comparisons are with differentiable methods such as NOTEARS, GOLEM-NV-$\ell_1$ and DAGMA​, rather than discrete methods like PC and FGES. Our results clearly demonstrate the advantages of CALM: it consistently outperforms other differentiable methods in all settings by increasing robustness to data standardization, addressing the inconsistency of $\ell_1$ penalties, achieving superior performance, and offering broader applicability to real-world problems. These key strengths underline CALM’s advantage as a more practical and reliable approach for differentiable structure learning.
> While CALM sometimes underperforms compared to PC and FGES in sparse graph settings, this is expected given the severe non-convexity challenges faced by differentiable methods. However, in dense graph settings, CALM even surpasses PC and FGES, showcasing its additional strengths and practicality. We will provide the discussion of CALM's advantages in the revision.
>
> **Q5** : “Minor Comment: i. Why are uppercase letters used for both matrices and vectors? ii. In line 97, X is $n \times d$, but B is $d \times d$, how do we compute $B^\top X$? iii. How is SHD computed? iv. What is PC method? What does PC stand for?”
>
> **A5** : Thank you for your comments. In our paper, uppercase letters are used for both variables $X$ (vectors in $\mathbb{R}^d$) and matrices $B$ to align with conventions in the literature on Structural Equation Models (SEMs). Specifically, $X = (X_1, \ldots, X_d)^\top$ represents a $d$-dimensional vector of random variables, and $B$ is a $d \times d$ coefficient matrix encoding the linear relationships between the variables. While the same notation is used, the context clarifies their roles. Regarding line 97, $X$ refers to the $d$-dimensional random variables ($X \in \mathbb{R}^d$) rather than the $n \times d$ data matrix. The model $X = B^\top X + N$ is expressed in terms of random variables, where $B^\top X$ is well-defined as $B^\top$ is $d \times d$, and $X$ is $d$-dimensional. For Structural Hamming Distance (SHD), it measures the number of edge additions, deletions, and reversals needed to transform the estimated graph into the ground truth graph. SHD is computed between the CPDAGs (Completed Partially Directed Acyclic Graphs) of the true and estimated graphs, accounting for equivalence classes. Lastly, the PC algorithm (Spirtes \& Glymour, 1991) is a discrete causal discovery method. It first uses conditional independence tests to prune edges from a fully connected graph, creating the skeleton, and then applies orientation rules to infer directions, producing a CPDAG that represents the Markov equivalence class of the true DAG.
>
> **References** :
>
> [1] Vowels, M. J., Camgoz, N. C., and Bowden, R. D’ya like DAGs? A survey on structure learning and causal discovery. ACM Computing Surveys, 55(4), nov 2022.
>
> [2] C. Glymour, K. Zhang, and P. Spirtes. Review of causal discovery methods based on graphical models. Frontiers in Genetics, 10, 2019.
>
> [3] Dong, X., Huang, B., Ng, I., Song, X., Zheng, Y., Jin, S., Legaspi, R., Spirtes, P., and Zhang, K. A versatile causal discovery framework to allow causally-related hidden variables. In International Conference on Learning Representations, 2024.
>
> [4] A. P. Singh and A. W. Moore. Finding optimal Bayesian networks by dynamic programming. Technical report, Carnegie Mellon University, 2005.

---

> ### Author Response · Authors · 2024-11-25
> **A Kind Request for Further Feedback**
>
> Dear Reviewer 1ULx,
>
> Thanks again for taking the time to review our work. We have carefully considered your comments and provided responses to them. Since the discussion period will end in 45 hours, we would like to kindly request for further feedback. Could you please check whether the responses properly addressed your concern? Thank you very much.
>
> Best regards,
>
> Authors of Paper 4754

---

> ### Comment · Reviewer_1ULx · 2024-11-25
>
> I thank the authors for their effort that they have put in for every responses to all reviewers, and for providing such detailed responses. I have raised my score to 5.
> However, given the amount of details that were missing in the paper, I am not sure the paper is above the threshold for acceptance, including the following reasons.
> 1. The computational cost versus performance trade offs between the \ell_0 based methods proposed versus existing methods is not clear. It is surprising that the \ell_0 solver can have similar cost as other differentiable structure learning methods. Is there no tradeoff?
> 2. The sparse versus dense graph performance comparison is really interesting. This is worth highlighting in the paper. But, aren't sparse DAGs more common in Bayesian networks, where other methods seem to be better. The penalty term (\ell_1 or \ell_0) is indeed used for promoting sparsity.

---

> > ### Author Response · Authors · 2024-11-30
> > **Thank you--did our response properly address your concerns?**
> >
> > Dear Reviewer 1ULx,
> >
> > We greatly appreciate your time and feedback. We have provided responses to your further comments. Could you please check whether the responses properly addressed your concerns? If you have any further questions or concerns, we hope to have the opportunity to address them. Your feedback would be appreciated. Thanks a lot for your time.
> >
> > Best regards,
> >
> > Authors of Paper 4754

---

> > ### Author Response · Authors · 2024-12-03
> > **A Kind Request for Feedback on our Further Response**
> >
> > Dear Reviewer 1ULx,
> >
> > We are very grateful for your further comments. At the same time, we are so eager to see whether your remaining concerns were properly addressed by our response that was carefully prepared--your feedback is so important to this work. If you have further questions or concerns, we will appreciate the opportunity to respond. Since the discussion period is ending in less than 10 hours, could you please kindly provide feedback on our further response?
> >
> > Many thanks for your time and effort,
> >
> > Authors of Paper 4754

---

> > > ### Comment · Reviewer_1ULx · 2024-12-03
> > >
> > > I thank the authors for the clarifications related to the follow up questions. The cost vs performance trade off is a key point and I recommend highlighting this. In order to not leave the scores ambiguous and to make it uniform, I have raised the score to 6. The initial submission was lacking wrt. a number of factors (several details were missing, numerical results were limited, etc) as highlighted by all reviewers. I commend the authors for their efforts during the discussion phase for all detailed responses, new experimental results, and revisions to the draft.

---

> > > > ### Author Response · Authors · 2024-12-03
> > > > **Acknowledgment for Reviewer 1ULx’s Feedback and Suggestions**
> > > >
> > > > Dear Reviewer 1ULx,
> > > >
> > > > We greatly appreciate your insightful feedback, time dedicated, and your recognition of our efforts during the discussion phase. Your constructive comments and suggestions have helped improve the manuscript. We will incorporate all suggestions in the revision.
> > > >
> > > > Best regards,
> > > >
> > > > Authors of Paper 4754

---

> ### Author Response · Authors · 2024-11-26
> **Follow-Up Response to Reviewer 1ULx**
>
> We sincerely appreciate your feedback on our responses and the time you have dedicated. Please find our detailed responses below:
>
> **Q1:** "computational cost versus performance trade offs between the $\ell_0$ based methods proposed versus existing methods is not clear" and "Is there no tradeoff?"
>
> **A1:** Thanks for asking this question. We would like to clarify that there is a trade-off between the $\ell_0$ based methods proposed versus existing methods. As mentioned in our response to Reviewer 2acW, while CALM does not demonstrate a runtime advantage over other differentiable methods on small and sparse graphs, it may even require less time than some alternatives on larger graphs. For instance, on 50-node SF4 graphs, CALM takes approximately 2500 seconds per run, compared to 20 seconds for NOTEARS and 150 seconds for GOLEM. However, on 200-node ER4 graphs, CALM takes around 4500 seconds per run, whereas NOTEARS takes about 6500 seconds and GOLEM approximately 3000 seconds.
>
> This demonstrates that while CALM’s runtime is not the fastest, it scales reasonably well with the graph size, and its performance does not degrade disproportionately as the number of nodes grows. Furthermore, CALM delivers superior results in terms of structural hamming distance (SHD) of CPDAG, skeleton precision, and skeleton recall, particularly for dense and large graphs. These significant performance improvements justify the additional computational cost.
>
> We believe this trade-off between runtime and performance is acceptable, given the substantial gains in accuracy compared to other differentiable methods. We hope this clarification addresses your concern. We have updated the manuscript to include this discussion of the tradeoff in Appendix H.
>
> **Q2:** "The sparse versus dense graph performance comparison is really interesting. This is worth highlighting in the paper." and "aren't sparse DAGs more common in Bayesian networks, where other methods seem to be better"
>
> **A2:** Thanks for your comment. Following your suggestion, we have updated the manuscript to highlight the performance comparison between sparse versus dense graph performance at the end of Section 4.6.
>
> Indeed, we agree with the reviewer that our proposed method may not outperform certain **non-differentiable baselines** on sparse graphs, specifically PC and FGES, but only outperforms them on dense graphs. At the same time, dense graphs are also rather common in Bayesian networks (e.g., many networks on the bnlearn data repository [1] are rather dense, including  Mehra, Sangiovese, Insurance and Water) and are typically considered more challenging. It is worth noting that both Reviwers viZA and 2acW also suggested us to include more experiments on dense graphs, which we provided in the responses and updated manuscript.
>
> Moreover, as emphasized in our response, our main contributions are to identify and analyze the inconsistency of the $\ell_1$ penalty for learning Markov equivalence classes, and accordingly investigate how to develop a differentiable approach that mitigates this issue, leading to a more practical and robust differentiable approach for learning Markov equivalence classes. As demonstrated in our experiments, our proposed method, CALM,  **outperforms existing differentiable methods** across all settings, including sparse and dense graphs.
>
> We have updated the manuscript to include the discussion above in Section 4.7. We hope that this addresses all your concerns, and want to thank the reviewer again for all the valuable feedback.
>
> ---
> **References:**
>
> [1] Scutari M (2010). “Learning Bayesian Networks with the bnlearn R Package.” Journal of Statistical Software, 35(3), 1–22.

---

### Official Review · Reviewer_2acW · 2024-11-03

**Soundness:** 2
**Presentation:** 2
**Contribution:** 2
**Rating:** 6
**Confidence:** 4

**Summary:**

The paper focuses on differentiable DAG learning in the linear Gaussian case. The author showed by examples that many adjacency matrices $B$ can generate the same covariance matrix as the ground truth adjacency matrix $B^*$ and satisfy $||B||_1<||B^*||_1$. Therefore, this raises doubts about the consistency of the $\ell_1$ regularization. To resolve the issue, they proposed CALM with an objective function that incorporates a Gumbel-Softmax mask and a moral graph mask to approximate the $\ell_0$ regularization. Empirical studies show the effectiveness of the method.

**Strengths:**

1. The moral graph mask is typically effective in the training.
2. The proposed method is robust to data normalization and outperforms the baseline differentiable learning methods using $\ell_1$ penalty.

**Weaknesses:**

1. I have some doubts about the motivating examples. In particular, if we want to show that Equation (1) can lead to inconsistent training results, then we need to identify the exact $B_{\min}$ that generates the same covariance matrix as $B^*$ with the minimal $\ell_1$ norm and prove that it is not equivalent to $B^*$. However, in Section 3.3, the author barely provides one example of the adjacency matrix that has a smaller $\ell_1$ norm and is not equivalent to $B^*$. It remains unclear whether the minima is equivalent to $B^*$. Therefore, it can not be regarded as a real counterexample. In Section 3.2, the author generates more examples using the Cholesky decomposition. I wonder whether it can generate all possible DAGs satisfying the covariance constraint. If so, then the justification is more convincing.
2. The novelty of the proposed method is limited. The Gumbel-Softmax mask, together with the $\ell_1$ penalty on $g_r(U)$ are also used similarly in Ng et al. (2022b) and in Equation (9) of Brouillard et al. (2020). The main difference in the object function seems to be the application of the moral graph mask.
3. The experiments are not sufficient, and the outperformance of CALM is not clearly established or understood. For example, on ER1, CALM and other differentiable DAG learning methods are generally worse than PC and FGES. Although CALM shows competitive results on standardized data with $n=1000$, this is not the case for $n=10e6$. Therefore, I would not be motivated to use differentiable methods in this setting. On ER4 with $d=50$ however, CALM shows clear outperformance, but the results for other $d$ are not known. Therefore, I encourage the author to conduct further experiments on dense graphs (ER2, ER4, SF2, SF4 with d = 20, 50, 70, 100) to illustrate the performance more clearly.
4. Other baselines can also be added, for example, DAGMA, which uses a different DAG constraint.

**Questions:**

1. What is the runtime of the proposed method compared to others?
2. Is $\tau$ a learned parameter or a tuning parameter?

---

> ### Author Response · Authors · 2024-11-23
> **Responses to Reviewer 2acW (part 1/5)**
>
> Thank you very much for your valuable comments and insightful suggestions. We have carefully considered your feedback and addressed each point below.
>
> **Q1** : “It remains unclear whether the minima is equivalent to $B^*$. Therefore, it can not be regarded as a real counterexample. In Section 3.2, the author generates more examples using the Cholesky decomposition. I wonder whether it can generate all possible DAGs satisfying the covariance constraint. ”
>
> **A1** : Thanks for this insightful question. In light of your question, we will include an additional proposition in the revision to show that the procedure described in Lines 200-206 (via permutations and Cholesky decomposition) can generate all possible DAGs satisfying the covariance constraint, which thus allows us to find the global minimum of Eq. (1). This is because each DAG corresponds to a causal ordering (i.e., permutation), and the parameters of such causal ordering that can generate the covariance constraint is unique and can be obtained via Cholesky decomposition, thanks to the property of acyclicity. This idea has been used in the sparsest permutation algorithm (Raskutti & Uhler, 2018). We will include a proposition and discussion to make this clear in the revision.
>
>
> **Q2**：“The novelty of the proposed method is limited. The Gumbel-Softmax mask, together with the $\ell_1$ penalty on $g_{\tau}(U)$ are also used similarly in Ng et al. (2022b) and in Equation (9) of Brouillard et al. (2020). ”
>
> **A2**: Thanks for highlighting this point. In our paper, we focus on identifying and analyzing the inconsistency of the $\ell_1$ penalty for learning Markov equivalence classes, and accordingly investigate how to develop a differentiable approach that mitigates this issue. Our primary goal is not to propose a completely novel method but to develop a more practical and robust differentiable approach for learning Markov equivalence classes. Our method offers better performance, increased robustness to data standardization, leading to potentially broader applicability to real-world problems. We will include this discussion in the revision and hope that it addresses your concern.

---

> ### Author Response · Authors · 2024-11-23
> **Responses to Reviewer 2acW (part 2/5)**
>
> **Q3**: “The experiments are not sufficient, and the outperformance of CALM is not clearly established or understood...I encourage the author to conduct further experiments on dense graphs (ER2, ER4, SF2, SF4 with d = 20, 50, 70, 100) to illustrate the performance more clearly.”
>
> **A3** : Thank you for your thoughtful comments. We will first clarify the outperformance of CALM and then present additional experiments on dense graphs to further illustrate its performance.
>
> (1) Clarification of CALM’s Outperformance: We would like to clarify that our work primarily focuses on developing a more practical differentiable method for learning Markov equivalence classes, as differentiable structure learning has recently gained significant attention in the research community. Consequently, our primary goal was to compare CALM with other differentiable methods, such as NOTEARS and GOLEM-NV-$\ell_1$​, and the results clearly demonstrate that CALM consistently outperforms these methods. Specifically, CALM exhibits increased robustness to data standardization, addresses the inconsistency of $\ell_1$​ penalties, and achieves superior performance across all evaluated settings. While we also included comparisons with discrete methods such as PC and FGES, it is important to note that these methods are not the primary focus of our work. Discrete methods like PC and FGES typically excel in sparse graph settings due to their fundamentally different optimization approaches that avoid the severe non-convexity challenges faced by differentiable methods. As such, it is not surprising that CALM may underperform compared to these methods in certain sparse scenarios. Nonetheless, CALM demonstrates clear superiority over other differentiable methods in all evaluated cases, particularly under data standardization. Moreover, in dense graph settings, CALM even surpasses discrete methods like PC and FGES, which is an additional strength of our approach. These results highlight CALM's robustness and practicality within the context of differentiable structure learning. We will include this clarification in the revision. We have also updated the manuscript to include the revisions and improvements in CALM over existing differentiable methods in Section 4.7.
>
> (2) Further experiments on dense graphs: We have conducted additional experiments to explore the performance of CALM on dense graphs, as you suggested. Specifically, we evaluated CALM and other baseline methods on 20-node ER4, 50-node SF4 (50-node ER4’s results have already been presented in initial submission), 70-node ER4, and 200-node ER4 graphs with 1000 samples under data standardization. (Due to time constraints, we could not cover all combinations, i.e., ER2, ER4, SF2, SF4 with d=20,50,70,100, but we will include the complete results in the revision.) Additionally, we included DAGMA in the baselines. The results are as follows:
>
> ***
>
> ### 20-node ER4 graphs (1000 Samples with Data Standardization)
>
> | Method         | SHD of CPDAG       | Precision of Skeleton | Recall of Skeleton |
> |----------------|--------------------|------------------------|--------------------|
> | CALM           | 64.3 ± 3.1        | 0.67 ± 0.03           | 0.64 ± 0.05       |
> | GOLEM-NV-$\ell_1$ | 85.7 ± 3.5        | 0.58 ± 0.03           | 0.51 ± 0.06       |
> | NOTEARS        | 85.3 ± 1.8        | 0.70 ± 0.05           | 0.20 ± 0.01       |
> | PC             | 81.3 ± 1.5        | 0.65 ± 0.02           | 0.25 ± 0.01       |
> | FGES           | 114.0 ± 8.5       | 0.48 ± 0.02           | 0.82 ± 0.02       |
> | DAGMA          | 92.7 ± 3.6        | 0.59 ± 0.04           | 0.38 ± 0.02       |
>
> ***
>
> ### 50-node SF4 graphs (1000 Samples with Data Standardization)
>
> | Method         | SHD of CPDAG       | Precision of Skeleton | Recall of Skeleton |
> |----------------|--------------------|------------------------|--------------------|
> | CALM           | 129.3 ± 22.4      | 0.68 ± 0.05           | 0.72 ± 0.03       |
> | GOLEM-NV-$\ell_1$ | 176.7 ± 2.1       | 0.80 ± 0.05           | 0.15 ± 0.03       |
> | NOTEARS        | 189.3 ± 1.1       | 0.77 ± 0.01           | 0.21 ± 0.03       |
> | PC             | 160.3 ± 4.3       | 0.91 ± 0.03           | 0.26 ± 0.01       |
> | FGES           | 157.7 ± 4.4       | 0.74 ± 0.04           | 0.74 ± 0.04       |
> | DAGMA          | 319.3 ± 3.8       | 0.38 ± 0.02           | 0.45 ± 0.02       |

---

> ### Author Response · Authors · 2024-11-23
> **Responses to Reviewer 2acW (part 3/5)**
>
> ***
>
> ### 70-node ER4 graphs (1000 Samples with Data Standardization)
>
> | Method         | SHD of CPDAG       | Precision of Skeleton | Recall of Skeleton |
> |----------------|--------------------|------------------------|--------------------|
> | CALM           | 180.7 ± 18.3      | 0.69 ± 0.03           | 0.78 ± 0.03       |
> | GOLEM-NV-$\ell_1$ | 291.0 ± 3.6       | 0.58 ± 0.03           | 0.19 ± 0.02       |
> | NOTEARS        | 291.0 ± 2.6       | 0.70 ± 0.04           | 0.15 ± 0.02       |
> | PC             | 287.0 ± 2.6       | 0.58 ± 0.02           | 0.21 ± 0.01       |
> | FGES           | 675.0 ± 28.5      | 0.29 ± 0.01           | 0.81 ± 0.03       |
> | DAGMA          | 338.0 ± 11.0      | 0.50 ± 0.01           | 0.33 ± 0.03       |
>
>
> ***
>
> ### 200-node ER4 graphs (1000 Samples with Data Standardization)
>
> | Method         | SHD of CPDAG       | Precision of Skeleton | Recall of Skeleton |
> |----------------|--------------------|------------------------|--------------------|
> | CALM           | 351.0 ± 67.1      | 0.77 ± 0.04           | 0.86 ± 0.03       |
> | GOLEM-NV-$\ell_1$ | 779.0 ± 9.4       | 0.78 ± 0.04           | 0.19 ± 0.04       |
> | NOTEARS        | 809.7 ± 16.3      | 0.72 ± 0.04           | 0.17 ± 0.01       |
> | PC             | 780.0 ± 10.6      | 0.60 ± 0.04           | 0.23 ± 0.02       |
> | FGES           | 1684.7 ± 205.9    | 0.31 ± 0.04           | 0.81 ± 0.02       |
> | DAGMA          | 921.7 ± 34.8      | 0.51 ± 0.04           | 0.32 ± 0.01       |
>
> ***
>
> From the results above, we observe that CALM consistently outperforms other methods, including PC and FGES, on these dense graphs across all metrics. This demonstrates the robustness and effectiveness of CALM even in challenging dense graph scenarios. We have updated the manuscript to include these results in Appendix E.

---

> ### Author Response · Authors · 2024-11-23
> **Responses to Reviewer 2acW (part 4/5)**
>
> **Q4** : “Other baselines can also be added, for example, DAGMA, which uses a different DAG constraint.”
>
> **A4** : Thank you for your suggestion. As mentioned in the response to Q3, we have already included DAGMA as a baseline in our new added experiments. In addition to the newly introduced graph types, we have also added DAGMA as a baseline for 50-node ER1, 50-node ER4, and 100-node ER1 graphs under both 1000 and $10^6$ samples with data standardization. The experimental settings, including running the experiments on the same 10 seeds, are consistent with those described in Table 5 of the initial submission. The results are summarized below:
>
> ***
>
> ### 50-node ER1 graphs
>
> | Method         | SHD of CPDAG (1000 Samples) | Precision of Skeleton (1000 Samples) | Recall of Skeleton (1000 Samples) | SHD of CPDAG ($10^6$ Samples) | Precision of Skeleton ($10^6$ Samples) | Recall of Skeleton ($10^6$ Samples) |
> |----------------|-----------------------------|---------------------------------------|-----------------------------------|--------------------------------|----------------------------------------|----------------------------------------|
> | CALM           | 12.1 ± 2.7                 | 0.93 ± 0.01                          | 0.98 ± 0.00                     | 7.0 ± 2.5                     | 0.97 ± 0.01                           | 0.98 ± 0.01                           |
> | DAGMA          | 73.3 ± 4.0                 | 0.57 ± 0.02                          | 0.95 ± 0.01                     | 70.6 ± 3.4                    | 0.59 ± 0.02                           | 0.95 ± 0.01                           |
>
> ***
>
> ### 50-node ER4 graphs
>
> | Method         | SHD of CPDAG (1000 Samples) | Precision of Skeleton (1000 Samples) | Recall of Skeleton (1000 Samples) | SHD of CPDAG ($10^6$ Samples) | Precision of Skeleton ($10^6$ Samples) | Recall of Skeleton ($10^6$ Samples) |
> |----------------|-----------------------------|---------------------------------------|-----------------------------------|--------------------------------|----------------------------------------|----------------------------------------|
> | CALM           | 168.8 ± 8.3                | 0.62 ± 0.02                          | 0.67 ± 0.02                     | 139.4 ± 10.2                  | 0.68 ± 0.02                           | 0.75 ± 0.02                           |
> | DAGMA          | 253.0 ± 7.5                | 0.49 ± 0.02                          | 0.33 ± 0.02                     | 252.5 ± 6.4                   | 0.49 ± 0.02                           | 0.33 ± 0.02                           |
>
> ***
>
> ### 100-node ER1 graphs
>
> | Method         | SHD of CPDAG (1000 Samples) | Precision of Skeleton (1000 Samples) | Recall of Skeleton (1000 Samples) | SHD of CPDAG ($10^6$ Samples) | Precision of Skeleton ($10^6$ Samples) | Recall of Skeleton ($10^6$ Samples) |
> |----------------|-----------------------------|---------------------------------------|-----------------------------------|--------------------------------|----------------------------------------|----------------------------------------|
> | CALM           | 26.7 ± 3.6                 | 0.90 ± 0.01                          | 0.99 ± 0.00                     | 17.0 ± 2.9                     | 0.96 ± 0.01                           | 0.99 ± 0.00                           |
> | DAGMA          | 152.6 ± 3.9                | 0.55 ± 0.01                          | 0.95 ± 0.01                     | 150.8 ± 3.3                    | 0.55 ± 0.01                           | 0.95 ± 0.01                           |
>
> ***
>
> It can be seen from the results that CALM demonstrates significant advantages over DAGMA across all scenarios. We have updated the manuscript’s Section 4.6 and Table 5 to include DAGMA's results. Furthermore, we have updated Appendix A.6 in the manuscript to include DAGMA's implementation details.

---

> ### Author Response · Authors · 2024-11-23
> **Responses to Reviewer 2acW (part 5/5)**
>
> **Q5** : “What is the runtime of the proposed method compared to others?”
>
> **A5** :   Thank you for raising this important question regarding runtime. While CALM does not demonstrate a runtime advantage over other differentiable methods on small and sparse graphs, it may even require less time than some alternatives on larger graphs. For instance, on 50-node SF4 graphs, CALM takes approximately 2500 seconds per run, compared to 20 seconds for NOTEARS and 150 seconds for GOLEM. However, on 200-node ER4 graphs, CALM takes around 4500 seconds per run, whereas NOTEARS takes about 6500 seconds and GOLEM approximately 3000 seconds.
>
> This demonstrates that while CALM’s runtime is not the fastest, it scales reasonably well with the graph size, and its performance does not degrade disproportionately as the number of nodes grows. Furthermore, CALM delivers superior results in terms of structural hamming distance (SHD) of CPDAG, skeleton precision, and skeleton recall, particularly for dense and large graphs. These significant performance improvements justify the additional computational cost.
>
> We believe this trade-off between runtime and performance is acceptable, given the substantial gains in accuracy compared to other differentiable methods. We hope this clarification addresses your concern. We will include this clarification in the revision.
>
> **Q6** : “Is $\tau$ a learned parameter or a tuning parameter?”
>
> **A6** : Thank you for your question regarding $\tau$. $\tau$ is a tuning parameter in our method. In the paper, we selected $\tau$ = 0.5 because our experiments showed that this value consistently led to excellent performance across various settings.

---

> > ### Comment · Reviewer_2acW · 2024-11-24
> > **Reply to the rebuttal**
> >
> > I thank the authors for their efforts in the rebuttal session. My concerns are well-addressed. I think the paper suggests a very interesting flaw in the current differentiable learning methods using the L1 regularization, which is valuable. In addition, the proposed algorithm showed strong performance on dense graphs which is generally the hardest setting for all methods. Therefore, I will raise my score to 6.

---

> > > ### Author Response · Authors · 2024-11-30
> > > **Acknowledgment for Reviewer 2acW’s Feedback and Suggestions**
> > >
> > > Dear Reviewer 2acW,
> > >
> > > We sincerely thank the reviewer for your constructive feedback, the time you have dedicated, and your recognition of our work. We will incorporate all the suggestions in the revised manuscript.
> > >
> > > Best regards,
> > >
> > > Authors of Paper 4754

---

### Official Review · Reviewer_viZA · 2024-11-04

**Soundness:** 3
**Presentation:** 4
**Contribution:** 3
**Rating:** 6
**Confidence:** 3

**Summary:**

This is an experimental paper that proposes a new algorithm CALM (Continuous and Acyclicity-constrained L0-penalized likelihood with estimated Moral graph) for causal structure learning of DAGs for the linear Gaussian case. The case where the covariance matrix of the noise is diagonal with equal variance of noise variables is well understood and can be solved by NoTears algorithm. The case of not equal variances is the main focus of this paper. Recent methods such as Golem propose minimizing L1-penalized likelihood function to recover unknown DAG structure.

This paper provides experimental evidence that L1 penalty may lead to a solution that is far from the true unknown DAG that minimizes L0 norm. It proposes a new algorithm that replaces L0 loss with Gumbel-Softmax to allow for smooth optimization techniques and provides an experimental evidence that the newly proposed method might outperform existing methods on sparse random DAGs (superior performance for ER4 graphs, though worse performance than PC for ER1 graphs). The paper also studies the effect of using Moral graph, data standardization and compares several L0 approximation methods.

**Strengths:**

The paper proposes a new algorithmic approach to an important problem of causal SEM learning for the case of linear Gaussian model. This problem is important for various practical applications and has received a lot of attention in the coming years. This paper tackles the challenging setup when noise covariances are unknown and might not be equal. This paper proposes to using  Gumbel-Softmax approximation to L0 loss with Moral graph and Data standardization instead of L1 penalty for solving SEM in case of linear Gaussian model. The authors provide an extensive study of the effects of each of the components of the algorithm: they compare several methods of L0 approximation and demonstrate the benefits of using Moral graph and Data Standardization.
The proposed algorithm is compared to several state of the art approaches (GOLEM, No Tears, PC, FGES) and authors demonstrate superiority of the proposed approach for a certain class of random graphs.

**Weaknesses:**

One of the weaknesses of the paper is that it does not discuss theoretical guarantees for the proposed methods and does not provide any set of conditions under which the method is guaranteed to find optimal or near-optimal solutions. I believe that some of such theoretical results should be possible to deduce from the existing results in the literature. Even though such guarantees might not be novel or go beyond existing results, I think that providing any such guarantees might be helpful to the reader.

Considering that this is an experimental paper I think it is beneficial to include experimental results on some real-world data to compare to existing methods.

**Questions:**

1. In Section 3, do I understand correctly that some of the d! permutations of B* could actually have a smaller L0 norm than B* itself? In which case B* is not actually an optimal solution? Can you please report what % of the initial graphs B* allow for smaller L0 norm among permuted matrices compared to B*.
2. I find it somewhat counterintuitive that 78% of permutations of the initial graph B* have a smaller L1 norm than graph B*. If B* is chosen truly at random I will expect that number to be around 50%. This seems to suggest that the observation authors make may be somewhat specific to the way the authors generate initial graph B*. Can you please provide a bit more intuition on how a number >50% makes sense in this setup?
3. Graphs ER1 and ER4 on 50 and 100 nodes are very sparse. Do you have an idea how well the proposed method performs in the regime when the unknown graph is less sparse, say ER4 on 15-20 nodes, or any other class of slightly more dense graphs?

---

> ### Author Response · Authors · 2024-11-23
> **Responses to Reviewer viZA (part 1/2)**
>
> Thank you very much for your valuable comments and insightful suggestions. We have carefully considered your feedback and addressed each point below.
>
> **Q1**: “One of the weaknesses of the paper is that it does not discuss theoretical guarantees for the proposed methods and does not provide any set of conditions under which the method is guaranteed to find optimal or near-optimal solutions.”
>
> **A1**: Thanks a lot for your thoughtful comments and suggestions. As you nicely suggested, the theoretical results can be derived from existing results in the literature. In light of your suggestion, we will include such results in the revision. Specifically, let $g_\tau(U)$ be a binary matrix and $\lambda_1=\log n / n$; under faithfulness assumption, the global minimum of CALM’s formulation (in Line 319) represents a structure that is Markov equivalent to the ground truth.
>
> **Q2**: “Considering that this is an experimental paper I think it is beneficial to include experimental results on some real-world data to compare to existing methods.”
>
> **A2**: Thank you for your valuable suggestion regarding evaluating our method on real-world datasets. We conducted experiments on the Sachs dataset [1], which is commonly utilized in probabilistic graphical model research to analyze the expression levels of proteins and phospholipids within human cells. The dataset contains d=11 variables and n=853 samples, with a ground truth of 17 edges. Our method, CALM, achieved an SHD of CPDAG of 12, outperforming GOLEM-NV-$\ell_1$ (SHD of CPDAG: 13) and NOTEARS (SHD of CPDAG: 22). These results demonstrate the strong performance of CALM on real-world data. We have updated the manuscript to include these results, along with the description, in Appendix G.
>
> **Q3**: “In Section 3, do I understand correctly that some of the d! permutations of $B^*$ could actually have a smaller  $\ell_0$ norm than $B^*$ itself? In which case $B^*$ is not actually an optimal solution?”
>
> **A3**: Thanks for asking this question. Under sparsest Markov faithfulness assumption, $B^*$ has the sparsest $\ell_0$ norm, and the only structures that have the same $\ell_0$ norm as $B^*$ are those Markov equivalent ones (Raskutti & Uhler, 2018). This has also been verified in the simulations. We will update the presentation of Section 3 and Table 1 in the revision to make this clear and avoid any possible confusion.
>
> **Q4**: “I find it somewhat counterintuitive that 78% of permutations of the initial graph $B^*$ have a smaller $\ell_1$ norm than graph $B^*$...Can you please provide a bit more intuition on how a number >50% makes sense in this setup?”
>
> **A4**: ​​Thanks for asking this question. Indeed, different settings/factors may affect the percentage of permutations with smaller $\ell_1$ norm than $B^*$, such as graph degree and graph type, as well as the range of edge weights and noise variances. In the simulations, we adopt the settings that are commonly used in existing works, such as Zheng et al. (2018), Ng et al. (2024).
>
> A possible intuitive explanation is that such a percentage may be related to varsortability (Reisach et al. 2021). For instance, consider two variable DAG $G^*:X_1\rightarrow X_2$ with linear coefficient $a$, as well as an alternative DAG $\hat{G}:X_1\leftarrow X_2$ with linear coefficient $b$. If $a$ is large and $\operatorname{Var}(X_2)$ is much larger than $\operatorname{Var}(X_1)$, then it is likely for $b$ to be considerably smaller than $a$, while the alternative DAG $\hat{G}$ can still generate the same covariance matrix. Studying the relationship between such percentage and varsortability may be an interesting direction for future works. We will include this discussion in the revision.

---

> ### Author Response · Authors · 2024-11-23
> **Responses to Reviewer viZA (part 2/2)**
>
> **Q5**: “Do you have an idea how well the proposed method performs in the regime when the unknown graph is less sparse, say ER4 on 15-20 nodes, or any other class of slightly more dense graphs?”
>
> **A5**: Thank you for raising the question regarding how the proposed method performs on less sparse graphs. To address this, we have conducted preliminary experiments and here are the results of 20-node ER4 graphs under 1000 samples with data standardization. The results are presented in the table below. We have updated the manuscript to include these results in Appendix E. We will include the final complete results in the revision.
>
> ***
>
> ### 20-node ER4 graphs (1000 Samples with Data Standardization)
>
> | Method         | SHD of CPDAG       | Precision of Skeleton | Recall of Skeleton |
> |----------------|--------------------|------------------------|--------------------|
> | CALM           | 64.3 ± 3.1        | 0.67 ± 0.03           | 0.64 ± 0.05       |
> | GOLEM-NV-$\ell_1$ | 85.7 ± 3.5        | 0.58 ± 0.03           | 0.51 ± 0.06       |
> | NOTEARS        | 85.3 ± 1.8        | 0.70 ± 0.05           | 0.20 ± 0.01       |
> | PC             | 81.3 ± 1.5        | 0.65 ± 0.02           | 0.25 ± 0.01       |
> | FGES           | 114.0 ± 8.5       | 0.48 ± 0.02           | 0.82 ± 0.02       |
> | DAGMA          | 92.7 ± 3.6        | 0.59 ± 0.04           | 0.38 ± 0.02       |
>
> ***
> From the results provided in the table, we observe that CALM achieves the best performance among all the methods, with the lowest SHD of CPDAG. This demonstrates that CALM remains effective even when the underlying graph is less sparse, as in the case of 20-node ER4 graphs.
>
> **References**:
>
> [1] K. Sachs, O. Perez, D. Pe’er, D. A. Lauffenburger, and G. P. Nolan. Causal protein-signaling networks
> derived from multiparameter single-cell data. Science, 308(5721):523–529, 2005.

---

> ### Author Response · Authors · 2024-11-25
> **A Kind Request for Further Feedback**
>
> Dear Reviewer viZA,
>
> Thanks again for taking the time to review our work. We have carefully considered your comments and provided responses to them. Since the discussion period will end in 45 hours, we would like to kindly request for further feedback. Could you please check whether the responses properly addressed your concern? Thank you very much.
>
> Best regards,
>
> Authors of Paper 4754

---

> ### Author Response · Authors · 2024-11-30
> **A Kind Request for Further Feedback**
>
> Dear Reviewer viZA,
>
> Thanks again for taking the time to review our work. We are writing to kindly let you know we have been eagerly looking forward to your feedback. If you have further questions or concerns after reading the rebuttal, we hope to have the opportunity to address them.
>
> With best regards,
>
> Authors of Paper 4754

---

### Meta-Review · Area_Chair_69X3 · 2024-12-11

**Metareview:**

This paper empirically shows that L1 penalized likelihood estimation for graph structured learning is inconsistent. It then proposes to use the L0 penalty and relaxes it to differentiable operators. On the positive side, reviewers appreciate the interesting finding of the inconsistency. On the other side, reviewers have concerns that no theoretical insights were drawn based on the experiments. Reviewers also identified that the novelty is limited, in that most proposed techniques have been known in the literature. While authors addressed many of the concerns during rebuttal, a major revision is required to place the paper above the bar. Therefore, the current version cannot be accepted.

**Additional Comments On Reviewer Discussion:**

Theoretical guarantees and technical novelty are two primary concerns from reviewers, along with a few questions regarding case study and experiments. While authors addressed many questions about experiments, the weakness in theory and technical contributions are still standing.

The AC initiated an AC-reviewer discussion and indicated that the above weakness will lead to reject, and indicated that reviewers have to justify if they would like to support the paper. There was no response from the reviewers.

The AC checked out all comments from the reviewers, and consider that this work would require a major revision in order to incorporate the feedback from the reviewers (which is also a point from Reviewer 1ULx).

---

### Decision · Program_Chairs · 2025-01-22

Reject